# CNS myelination and remyelination depend on fatty acid synthesis by oligodendrocytes

**Penelope Dimas[1†], Laura Montani[1†]\*, Jorge A Pereira[1], Daniel Moreno[1], Martin Trötzmüller[2], Joanne Gerber[1], Clay F Semenkovich[3], Harald C Köfeler[2], Ueli Suter[1]\***

[1]Institute of Molecular Health Sciences, Swiss Federal Institute of Technology, ETH Zürich, Zürich, Switzerland; [2]Center for Medical Research, Medical University of Graz, Graz, Austria; [3]Division of Endocrinology, Metabolism and Lipid Research, Washington University Medical School, St. Louis, United States

**\*For correspondence:**
laura.montani@biol.ethz.ch (LM);
ueli.suter@biol.ethz.ch (US)

[†]These authors contributed equally to this work

**Competing interests:** The authors declare that no competing interests exist.

**Abstract** Oligodendrocytes (OLs) support neurons and signal transmission in the central nervous system (CNS) by enwrapping axons with myelin, a lipid-rich membrane structure. We addressed the significance of fatty acid (FA) synthesis in OLs by depleting FA synthase (FASN) from OL progenitor cells (OPCs) in transgenic mice. While we detected no effects in proliferation and differentiation along the postnatal OL lineage, we found that FASN is essential for accurate myelination, including myelin growth. Increasing dietary lipid intake could partially compensate for the FASN deficiency. Furthermore, FASN contributes to correct myelin lipid composition and stability of myelinated axons. Moreover, we depleted FASN specifically in adult OPCs to examine its relevance for remyelination. Applying lysolecithin-induced focal demyelinating spinal cord lesions, we found that FA synthesis is essential to sustain adult OPC-derived OLs and efficient remyelination. We conclude that FA synthesis in OLs plays key roles in CNS myelination and remyelination.
DOI: https://doi.org/10.7554/eLife.44702.001

## Introduction

Under the control of a complex bidirectional signaling program (*Herbert and Monk, 2017*; *Klingseisen and Lyons, 2018*; *Nave and Werner, 2014*; *Osso and Chan, 2017*), oligodendrocytes (OLs) encase central nervous system (CNS) axons with myelin, a highly organized and compacted multi-membrane structure. Myelination allows rapid transmission of action potentials and preserves axonal integrity by multiple mechanisms, including metabolic support by OLs (*Fünfschilling et al., 2012*; *Lee et al., 2012*; *Simons et al., 2014*). Thus, loss of myelin in diseases of diverse etiology, such as in multiple sclerosis, ultimately results in axonal degeneration and clinical deterioration of affected patients (*Franklin and Ffrench-Constant, 2017*; *Nave and Werner, 2014*; *Saab and Nave, 2017*). During postnatal development, OL progenitor cells (OPCs) undergo a tightly orchestrated differentiation program leading to mature OLs that contact axons and wrap them with myelin (*Nave and Werner, 2014*). A significant population of OPCs remains resident in the adult CNS, designated adult OPCs (aOPCs) (*Dimou and Gallo, 2015*; *Franklin and Ffrench-Constant, 2017*). Upon a demyelinating injury, aOPCs are activated, migrate to the lesion site, proliferate, undergo differentiation, and mature into OLs that remyelinate axons. Hence, aOPCs-derived OLs substitute for preexisting ones which are lost upon demyelination (*Crawford et al., 2016*; *Franklin and Ffrench-Constant, 2017*). However, this program often fails in myelin-defective lesions leaving axons particularly vulnerable. While a number of exogenous and endogenous factors are known to restrict the

regenerative potential of aOPCs (*Franklin and Ffrench-Constant, 2017*; *Miron, 2017*), the roles of modulators of their metabolic state remain largely unknown.

Myelin is characterized by an exceptionally high lipid content (~80% of dry weight) (*Chrast et al., 2011*; *Nave and Werner, 2014*; *Schmitt et al., 2015*). Fatty acids (FAs) are fundamental building blocks for both glycolipids and phospholipids, which comprise the largest proportion of myelin membrane lipids (*Harayama and Riezman, 2018*). FAs can be acquired from the pool present in the circulation due to dietary intake (essential and non-essential FAs), through horizontal flux from adjacent cells (essential and non-essential FAs), or are cell-endogenously synthesized (non-essential FAs) (*Camargo et al., 2017*; *Currie et al., 2013*). However, the relative contribution of uptake versus synthesis to the final pool of FAs in OLs at different stages of differentiation remains to be determined.

The mTORC1-SCAP signaling axis regulates a plethora of lipogenic pathways via cleavage of Sterol Regulatory Element-Binding Proteins (SREBPs). These transcription factors are major modulators of FA and cholesterol metabolism (*Laplante and Sabatini, 2012*; *Porstmann et al., 2008*), including in OLs (*Camargo et al., 2017*). Accordingly, depletion of mTOR (*Wahl et al., 2014*), the functionally required mTORC1 subunit Raptor (*Bercury et al., 2014*; *Lebrun-Julien et al., 2014*) or SCAP (*Camargo et al., 2017*) in OLs causes reduced radial growth of myelin, mimicking the phenotype triggered by depletion of the SREBP-downstream target squalene synthase, an enzyme required for cholesterol synthesis (*Saher et al., 2005*). Genetic depletion of Raptor and SCAP in OLs results also in reduced expression of FA synthase (FASN) (*Camargo et al., 2017*; *Lebrun-Julien et al., 2014*) – the enzyme responsible for the synthesis of the 16-carbon palmitic acid that is used as substrate for subsequent synthesis of longer FAs (*Currie et al., 2013*). However, the functional role of endogenous FA synthesis in OL differentiation and CNS myelination, and how diminished FASN expression impacts upon the observed mTORC1/SCAP-knockout phenotypes, is not clear.

We previously found that de novo FA synthesis is essential for correct onset of myelination by Schwann cells, the myelinating cells of the peripheral nervous system (PNS) (*Montani et al., 2018*). In the present study, we have tested whether OLs rely also on endogenous FA synthesis for developmental myelination. Furthermore, we investigated whether remyelination after injury depends on FA synthesis by OLs. In development, we addressed this question by genetically depleting FASN in the OL lineage of transgenic mice. To analyze the role of endogenous FA synthesis in remyelination of adults, we deleted FASN specifically from aOPCs and their progeny, in conjunction with a focal demyelinating spinal cord lesion using the well-established lysolecithin gliotoxin-injection model (*Blakemore and Franklin, 2008*). Our data reveal that endogenous FA synthesis in OLs is required for correct CNS myelination in development and essential for efficient remyelination in adulthood.

## Results

### Depletion of FASN in oligodendrocytes

OLs have the capacity to cover numerous axons with multiple membrane layers to form myelin (*Nave and Werner, 2014*). Thus, we hypothesized that OLs may depend on endogenous FASN-driven FA synthesis for myelin production. Indeed, OLs transcribe FASN at high levels (http://web.stanford.edu/group/barres_lab/cgi-bin/igv_cgi_2.py?lname=Fasn) (*Zhang et al., 2014*). To test our hypothesis, we used mouse genetics to ablate FASN specifically in the OL lineage. Conditionally mutant $Fasn^{lox/lox}$ mice (*Chakravarthy et al., 2005*) were crossed with mice expressing Cre recombinase under the control of *Olig2* gene regulatory elements (*Schüller et al., 2008*) (*Figure 1a*). The resulting *Fasn* mutants were viable, fertile and born according to Mendelian distribution. Loss of FASN expression was confirmed by immunohistochemistry in differentiated (CC1+) OLs of postnatal day (P) 14 mutants in spinal cord white matter (*Figure 1b,c*), in agreement with highly efficient recombination in OL lineage cells as monitored by inclusion of the Cre-dependent reporter *Rosa26-loxPstoploxP-YFP* allele (*Srinivas et al., 2001*) (*Figure 1—figure supplement 1a,b*). Comparable loss of FASN was also found in the gray matter (*Figure 1—figure supplement 2a,b*). Note that large motoneurons in the ventral horn showed strong FASN expression, which was retained in mutant mice (*Figure 1—figure supplement 2a,b*).

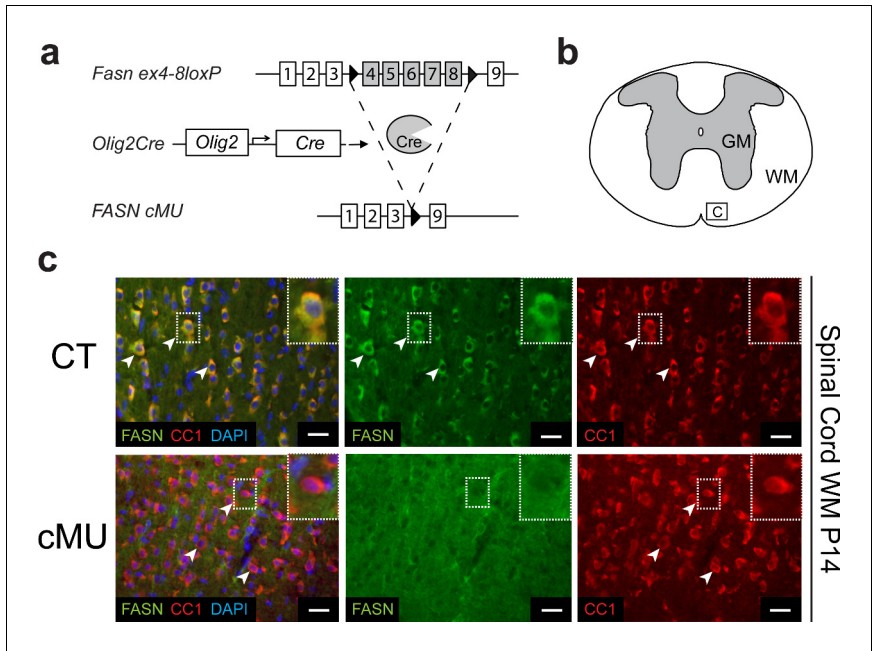

**Figure 1.** Generation of mutant mice lacking FASN in the oligodendrocyte lineage. (**a**) Experimental strategy of conditional *Fasn* allele inactivation upon *Olig2*-driven Cre expression in vivo. (**b**) Schematic of thoracic spinal cord. Square insert 'c': White matter (WM) where subsequent immunostaining images were acquired. GM = gray matter (**c**) Representative immunostaining of cross-sectioned spinal cord from P14 CT and cMU mice, *n* = 3 mice examined for each, CT and cMU. Prominent cytoplasmic FASN expression in differentiated OLs (CC1+; examples indicated by arrowheads) in CT, but not in cMU. Nuclear marker: DAPI. Scale bars: 20 μm. CT = control, cMU = conditional mutant, GM = gray matter, WM = white matter.
DOI: https://doi.org/10.7554/eLife.44702.002

The following figure supplements are available for figure 1:

**Figure supplement 1.** Recombination efficiency in spinal cord white matter.
DOI: https://doi.org/10.7554/eLife.44702.003

**Figure supplement 2.** FASN expression in gray matter oligodendrocytes of CT but not in cMU mice.
DOI: https://doi.org/10.7554/eLife.44702.004

## FASN is dispensable for proliferation and differentiation along the oligodendrocyte lineage

Before OLs reach the myelinating stage, their progenitors (OPCs) need to proliferate, differentiate and mature. In some proliferating cells such as in cancer and neuronal progenitors, FASN activity is required to support increased metabolic demands (*Currie et al., 2013*; *Knobloch et al., 2013*). Thus, we analyzed OL proliferation by EdU incorporation but found no detectable differences between mutants and controls at both P4 and P10 (*Figure 2a,b*). Next, we addressed whether lack of FASN may affect differentiation along the OL lineage during development. Initial immunohisto-chemical analyses revealed marginal (if any) FASN expression in white matter OPCs (PDGFRα+ cells) of the P14 spinal cord (*Figure 2—figure supplement 1a*), compared to high levels in differentiated OLs (CC1+ cells) (*Figure 2—figure supplement 1a*). Further comparative analyses of mutant and control mice yielded no significant differences in the percentages of OPCs (PDGFRα+ Olig2+) (*Figure 2—figure supplement 1b,c*), total OL lineage cells (Olig2+) (*Figure 2c,d and e*) and differentiated OLs (CC1+ Olig2+) (*Figure 2c,d and e*) in spinal cord white matter at P4, P10 and P14. Taken together, these data indicate that FASN is not required for correct proliferation and differentiation of the OL lineage to CC1+ cells during postnatal development.

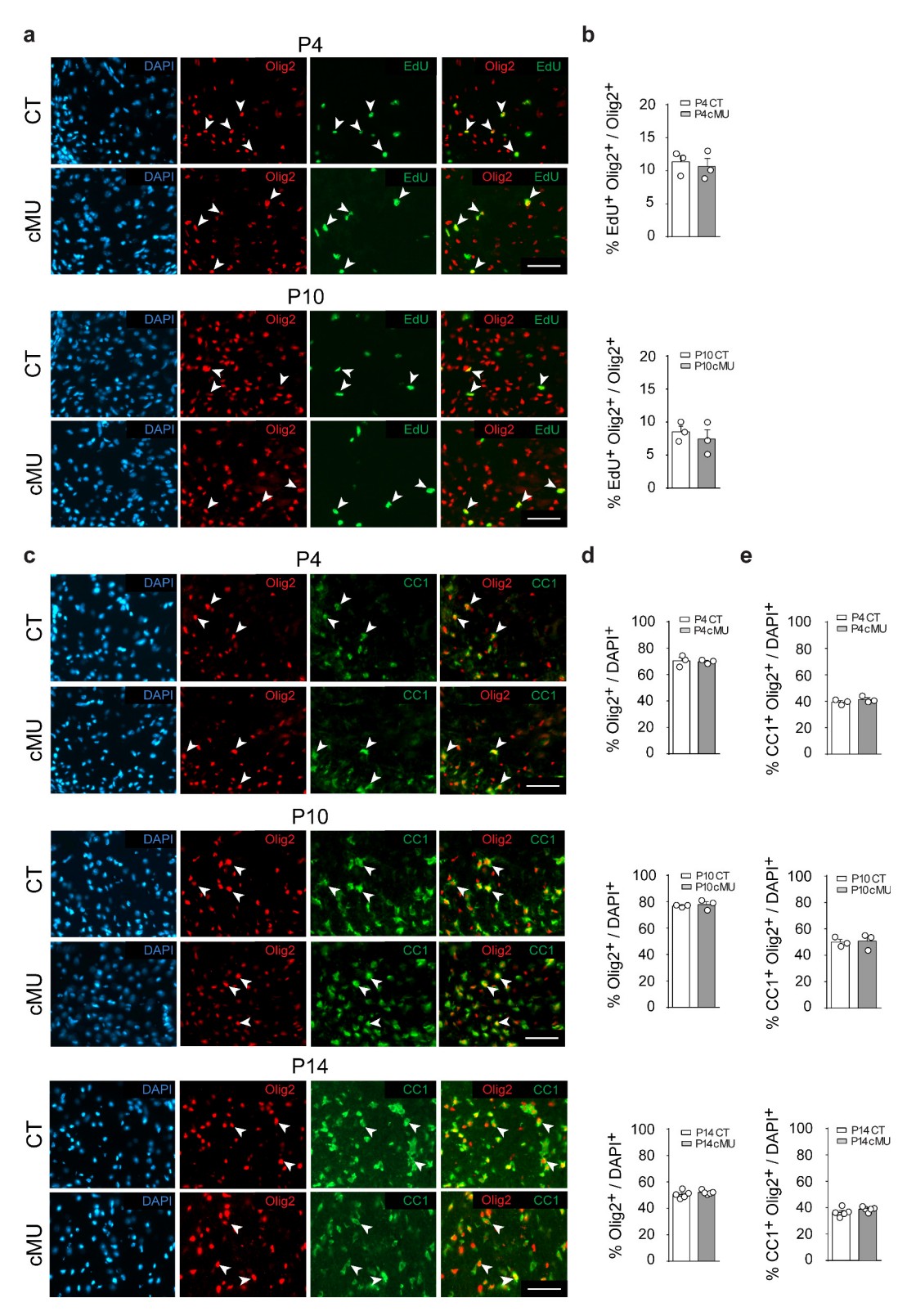

**Figure 2.** De novo fatty acid synthesis is not required to achieve correct numbers of oligodendrocyte lineage cells during development. (a) Representative immunostaining of ventral white matter in cross-sectioned spinal cords of P4 and P10 mice identifying proliferating (EdU+) OLs (Olig2+; examples indicated by arrowheads), $n$ = 3 mice for each, CT and cMU. Nuclear marker: DAPI. Scale bar: 50 µm, applying to entire panel. (b) Corresponding graphs to (a) with quantification of percentage of proliferating OLs (EdU+ Olig2+) over total number of OLs (Olig2+) in spinal cord

*Figure 2 continued on next page*

*Figure 2 continued*

white matter of CT and cMU mice at P4 and P10. Data points represent $n$ = 3 mice for each, CT and cMU; random fields of both dorsal and ventral white matter of 4 sections quantified per animal, with at least 83 Olig2+ cells quantified per section (unpaired two-tailed two sample Student's t-test; at P4: cMU vs. CT, p=0.6876, $t$ = 0.4326; P10: cMU vs. CT, p=0.5448, $t$ = 0.6608). (c) Representative immunostaining of ventral white matter in cross-sectioned spinal cords from P4, P10, and P14 mice identifying differentiated OLs (CC1+; examples indicated by arrowheads), OLs (Olig2+) and total number of cells (DAPI+), $n$ = 3 mice for each, CT and cMU at P4 and P10, $n$ = 5 mice for each, CT and cMU at P14. Nuclear marker: DAPI. Scale bar: 50 μm, applying to entire panel. (d, e) Corresponding graphs to (c) with quantification of percentage of total OLs (Olig2+) (d) and differentiated OLs (CC1 + Olig2+) (e) over total number of cells (DAPI+), in the spinal cord white matter of CT and cMU mice at P4, P10 and P14. Data points represent $n$ = 3 mice for each, CT and cMU at P4 and P10, and $n$ = 5 mice for each, CT and cMU at P14. Random fields of both dorsal and ventral white matter of at least 3 sections quantified per animal, with at least 83 Olig2+ cells quantified per section (unpaired two-tailed two sample Student's t-test; % Olig2+/ DAPI+ at P4: cMU vs. CT, p=0.8280, $t$ = 0.2319; at P10: cMU vs. CT, p=0.6694, $t$ = 0.46; at P14: cMU vs. CT, p=0.3340, $t$ = 1.028; % CC1+ Olig2+ / DAPI + at P4: cMU vs. CT, p=0.3405, $t$ = 1.081; at P10: cMU vs. CT, p=0.8490, $t$ = 0.2031; at P14: cMU vs. CT, p=0.2061, $t$ = 1.376). Bars represent mean ±SEM. CT = control, cMU = conditional mutant.

DOI: https://doi.org/10.7554/eLife.44702.005

The following figure supplement is available for figure 2:

**Figure supplement 1.** FASN expression in oligodendrocyte progenitors is marginal and dispensable for their early maintenance.

DOI: https://doi.org/10.7554/eLife.44702.006

## De novo fatty acid synthesis is critical for accurate CNS myelination, including radial myelin growth

Based on the observed high levels of FASN expression in differentiated (CC1+) OLs, we addressed next whether these cells rely on endogenous FA synthesis to myelinate. Thus, we performed EM ultrastructural morphological analysis of the ventral spinal cord white matter from control and mutant mice at P14 (*Figure 3*). We found that mutant mice exhibited more axons not encased by myelin compared to controls (69.57 ± 3.75% in mutants vs. 53.56 ± 2.37% in controls) (*Figure 3a,b*). Moreover, g-ratio analysis (i.e. ratio of axon to fiber (axon + myelin) diameter) revealed that mutant myelinated axons had overall thinner myelin compared to controls (*Figure 3c*). These data were confirmed by analysis of the correlation of g-ratio and fiber diameter with axon diameter (*Figure 3d,e*). To understand whether lack of FASN resulted in persistently reduced myelination of axons, we compared the ultrastructural morphology of white matter of the ventral spinal cord of mutant versus control mice at P180. Although myelination progressed in both cases, a higher percentage of not-yet myelinated axons persisted in the spinal cord white matter of mutant mice (15.21 ± 2.92% in mutants vs. 6.59 ± 1.01% in controls) (*Figure 3g*). In addition, marked hypomyelination of mutant axons also persisted (*Figure 3f*, arrows), confirmed by g-ratio analysis (*Figure 3h,i and j*). We conclude that endogenous FA synthesis in OLs is critical for accurate myelination in the spinal cord, including efficient myelin growth. Our findings may reflect that lack of endogenous FA synthesis limits total myelin production of each mutant OL and/or the number of myelin sheaths that can be generated. Other contributing factors to the mutant OL phenotype might include altered timing in the onset of individual myelin sheath formation.

To examine whether anatomically differently located OLs are similarly dependent on endogenous FA synthesis, we extended our study to the optic nerve. As observed in the spinal cord, high recombination efficiency in OL lineage cells was also found in this structure (*Figure 3—figure supplement 1a,b*). In line with our findings in the spinal cord, P14 optic nerves of mutant mice displayed a higher percentage of axons not-yet encased by myelin compared to controls (84.08 ± 0.33% in mutants vs. 75.68 ± 0.77% in controls) (*Figure 3—figure supplement 2a,b*). Although the onset of myelination progressed in both mutants and controls, also P180 mutant optic nerves retained more axons not encased by myelin (21.16 ± 5.1% in mutants vs. 4.29 ± 1.6% in controls) (*Figure 3—figure supplement 2f*, false colored, and g). Overall, no significant hypomyelination was detectable in optic nerves of mutant mice in early development (P14) (*Figure 3—figure supplement 2c,d and e*). Analysis of P180 optic nerves revealed, however, that mutant OLs failed to support ongoing myelin growth accurately, resulting in pronounced hypomyelination in adulthood (*Figure 3—figure supplement 2f*). This observation was confirmed by overall g-ratio quantification (*Figure 3—figure supplement 2h*) and correlation analyses of g-ratio and fiber diameter versus axon diameter (*Figure 3—figure supplement 2i,j*). We conclude that FASN activity in OLs is required for accurate myelination also in the optic nerve, albeit with subtle differences to the spinal cord.

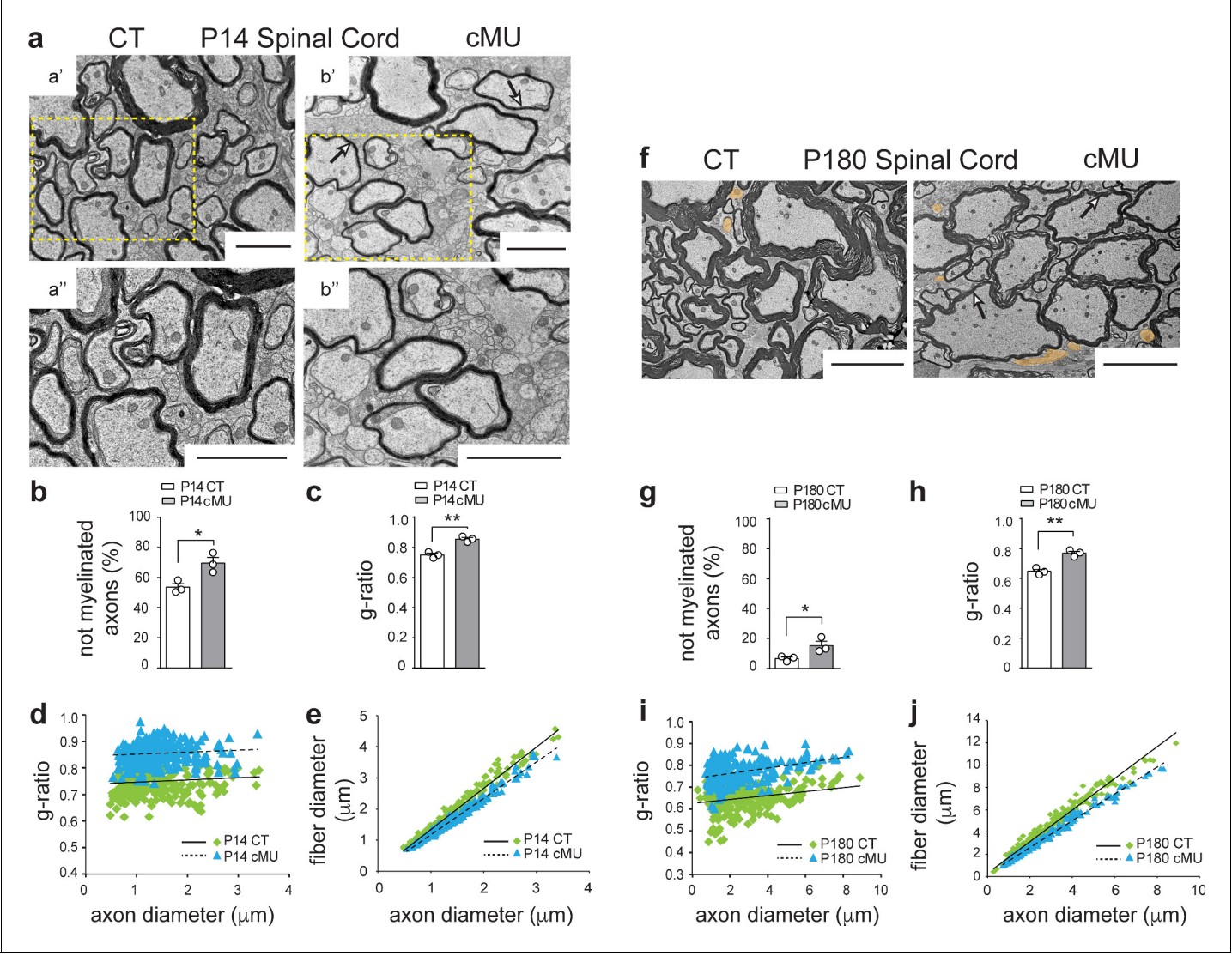

**Figure 3.** De novo fatty acid synthesis by oligodendrocytes is essential to achieve accurate myelination in the spinal cord. (a) Representative EM images of P14 control (CT, **a' and a''**) and conditional mutant (cMU, **b' and b''**) white matter from the ventral funiculi of lumbar spinal cords. cMUs show more naked axons, not-yet enwrapped by myelin, when compared to CTs. cMUs also display thinner myelin (examples indicated by arrows). Scale bars: 2 μm. (b) Corresponding graph with quantification of percentage of not myelinated axons at P14. Data points represent *n* = 3 mice for each, CT and cMU, with at least 590 axons quantified per animal in random fields selected in the same anatomical area (unpaired two-tailed two sample Student's t-test; at P14: cMU vs. CT, p=0.0225, *t* = 3.614), *p<0.05. (c) Overall hypomyelination at P14 in cMU as shown by g-ratio analysis. Data points represent *n* = 3 mice for each, CT and cMU (unpaired two-tailed two sample Student's t-test; at P14: cMU vs. CT, p=0.0025, *t* = 6.731), **p<0.01. (d, e) Linear correlation of g-ratio versus axon diameter (d) and of fiber diameter versus axon diameter (e), in the ventral white matter spinal cord of cMU compared to CT at P14. 100 myelinated axons per mouse analyzed in random fields selected in the same anatomical area, *n* = 3 mice for each, CT and cMU. (f) Representative EM images of P180 CT and cMU white matter from ventral funiculi of lumbar spinal cords. cMUs show more naked axons (false colored in orange) compared to CTs. cMU axons are encased by thinner myelin (examples indicated by arrows) compared to CTs. Scale bars: 5 μm. (g) Corresponding graph with quantification of percentage of not myelinated axons at P180. Data points represent *n* = 3 mice for each, CT and cMU, with at least 220 axons quantified per animal, in random fields selected in the same anatomical area (unpaired two-tailed two sample Student's t-test; at P180: cMU vs. CT, p=0.0493, *t* = 2.791), *p<0.05. (h) Overall hypomyelination at P180 in cMU compared to CT, as shown by g-ratio analysis. Data points represent *n* = 3 mice for each, CT and cMU (unpaired two-tailed two sample Student's t-test; at P180: cMU vs. CT, p=0.0027, *t* = 6.651), **p<0.01. (i, j) Linear correlation of g-ratio versus axon diameter (i) and of fiber diameter versus axon diameter (j), in the ventral white matter spinal cord of cMU compared to CT at P180. At least 65 myelinated axons per mouse analyzed in random fields selected in the same anatomical area, *n* = 3 mice for each, CT and cMU. Bars represent mean ±SEM. CT = control, cMU = conditional mutant.

DOI: https://doi.org/10.7554/eLife.44702.007

The following figure supplements are available for figure 3:

**Figure supplement 1.** Recombination efficiency in the optic nerve.

*Figure 3 continued on next page*

*Figure 3 continued*

DOI: https://doi.org/10.7554/eLife.44702.008

**Figure supplement 2.** De novo fatty acid synthesis by oligodendrocytes is essential to achieve accurate myelination of the optic nerve.

DOI: https://doi.org/10.7554/eLife.44702.009

**Figure supplement 3.** De novo fatty acid synthesis by oligodendrocytes is essential to achieve accurate myelination of the corpus callosum.

DOI: https://doi.org/10.7554/eLife.44702.010

Since the corpus callosum is often explored in myelin research, we examined also this brain region in the adult. Similar to our findings in the spinal cord and in the optic nerve, an increased fraction of axons that were not enwrapped by myelin was detected in the corpus callosum of P180 mutant mice (72.27 ± 0.98% in mutants vs. 63.53 ± 2.41% in controls) (*Figure 3—figure supplement 3a,b*). In addition, myelinated axons displayed relatively thinner myelin in mutants, as indicated by the higher g-ratio values compared to controls (*Figure 3—figure supplement 3a,c*) and confirmed by scatter plots of g-ratio and fiber diameter versus axonal diameter (*Figure 3—figure supplement 3d,e*).

Taken together, our studies of different CNS regions revealed that endogenous FA synthesis by OLs is required for correct myelination in various areas. More detailed comparisons require further investigations, including complementary approaches to alleviate technical limitations. Such examinations may well uncover more subtle variations between these regions, possibly relating to the physiological context of each environment, including local differences in astrocyte coupling, or potential intrinsic differences between oligodendrocyte lineage cells (*Dimou and Simons, 2017*; *Marques et al., 2016*).

## De novo fatty acid synthesis contributes to the stability of myelinated axons and correct CNS myelin lipid composition

A further characteristic feature in spinal cord tissue of P14 mutant mice were some aberrant vacuolated myelin-axon profiles (*Figure 4a*), confirmed by quantification (3.81 ± 0.15% in mutants vs. 0.25 ± 0.08% in controls) (*Figure 4b*). These structures resembled those observed in the PNS when Schwann cells lack FASN (*Montani et al., 2018*). Irregular myelin (*Figure 4a*, arrowheads) was often surrounding axons which appeared compressed (*Figure 4a*, asterisks). Similarly, examination of optic nerves revealed a comparable increase in anomalous myelin-axon profiles in mutant mice (5.41 ± 1.0% in mutants vs. 0.75 ± 0.19% in controls) (*Figure 4—figure supplement 1a,b*).

Since FAs are key structural components of cellular membranes, we investigated the lipid composition of myelin in mutant mice that lack FA synthesis in OLs compared to controls. To this end, we quantified FAs and FA-derived lipid classes in myelin purified from spinal cords of P60 mice. No significant alterations were detectable in essential (diet-derived), conditional-essential (derived from essential) and non-essential FAs (*Figure 4c*, *Figure 4—figure supplement 2a*) of mutants. Furthermore, we found no difference in the content of palmitate, the direct product of FASN enzymatic activity in mutant vs. control myelin (*Figure 4d*). Myelin from spinal cords of mutant mice contained also similar amounts of ceramides (*Figure 4e*, *Figure 4—figure supplement 2b*), cerebrosides (*Figure 4f*, *Figure 4—figure supplement 2c*), and sphingomyelin (*Figure 4g*, *Figure 4—figure supplement 2d*) compared to controls. Besides some minor alterations (*Figure 4—figure supplement 2*), strikingly elevated total levels of phosphatidylserines were found in mutants compared to controls (393.5 ± 25.53 in mutants vs. 269.6 ± 4.38 in controls, pmol/mg of proteins) (*Figure 4i*, *Figure 4—figure supplement 2g*). We conclude that lack of endogenous FA synthesis in OLs has a modest influence on the resulting lipid composition of myelin.

## Transcriptome analysis of optic nerves of *Fasn* mutant mice reveals oligodendrocyte defects in late stages of maturation, including myelination

Our results obtained so far indicated that FASN is critical for timely onset of OLs myelination and subsequent radial growth of CNS myelin, but does not play a major role in OL lineage progression based on expression of Olig2, PDGFRα, and CC1 as analytic markers. It is well known, however, that aside from being building blocks for different lipid species, FAs play other key roles such as a source

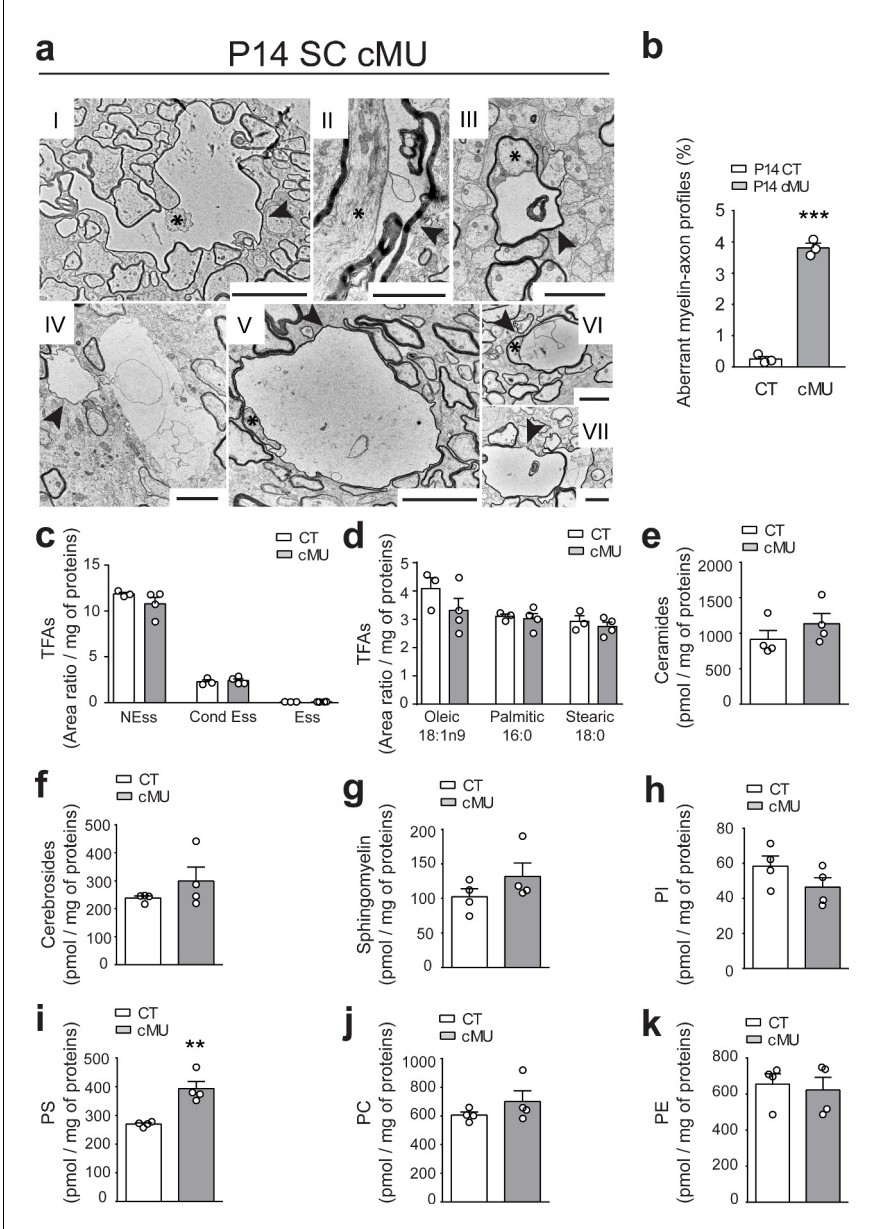

**Figure 4.** De novo fatty acid synthesis by oligodendrocytes is required to maintain structural stability of myelinated axons. (a) Exemplary EM images of aberrant myelin-axon profiles showing vacuolation (examples indicated by arrowheads) in the ventral white matter of spinal cords of P14 cMUs. Where identifiable, axons associated with aberrant structures are indicated by asterisks. Scale bars: I = 5 μm, II = 2 μm, III = 2 μm, IV = 2 μm, V = 5 μm, VI = 2 μm, VII = 2 μm. (b) Corresponding graph with quantification of the percentage of aberrant myelin-axon profiles. Data points represent $n$ = 3 mice for each, CT and cMU (unpaired two-tailed two sample Student's t-test; at P14: cMU vs. CT p<0.0001, $t$ = 21.16), ***p<0.001. At least 8300 axons per mouse analyzed in random fields selected in the same anatomical area. (c) Total FA composition of myelin purified from spinal cords of P60 CT and cMU mice, divided in non-essential FAs (NEss), conditional-essential FAs (Cond Ess) and essential FAs (Ess). Data points represent $n$ = 3 mice for CT and $n$ = 4 mice for cMU (unpaired two-tailed two sample Student's t-test; NEss: cMU vs. CT, p=0.2609, $t$ = 1.267; Cond Ess: cMU vs. CT, p=0.6625, $t$ = 0.4635; Ess: cMU vs. CT, p=0.4435, $t$ = 0.8317). (d) Quantification of the most abundant FAs found in myelin purified from spinal cords of CT and cMU mice. Data points represent $n$ = 3 mice for CT and $n$ = 4 mice for cMU (unpaired two-tailed two sample Student's t-test; Oleic: cMU vs. CT, p=0.2525, $t$ = 1.293; Palmitic: cMU vs. CT, p=0.7377, $t$ = 0.3542; Stearic: cMU vs. CT, p=0.4966, $t$ = 0.7328). (e, f, g) Content of ceramides (e), cerebrosides (f) and sphingomyelin (g) in myelin isolated from spinal cords of cMU compared to CT mice. Data points represent $n$ = 4 mice for each, CT and cMU (unpaired two-tailed two sample Student's t-test; ceramides: cMU vs. CT, p=0.2822, $t$ = 1.181;

*Figure 4 continued on next page*

*Figure 4 continued*

cerebrosides: cMU vs. CT, p=0.2722, *t* = 1.209; sphingomyelin: cMU vs. CT, p=0.2310, *t* = 1.333). (h, i, j, k) Total content of phosphatidylinositols (PI), phosphatidylserines (PS), phosphatidylcholines (PC) and phosphatidylethanolamines (PE) in myelin isolated from spinal cords of CT and cMU mice. Data points represent *n* = 4 mice for each, CT and cMU (unpaired two-tailed two sample Student's t-test; PI: cMU vs. CT, p=0.1765, *t* = 1.532; PS: cMU vs. CT, p=0.0031, *t* = 4.784; PC: cMU vs. CT, p=0.2698, *t* = 1.216; PE: cMU vs. CT, p=0.7236, *t* = 0.3706), **p<0.01. Bars represent mean ±SEM. CT = control, cMU = conditional mutant.

DOI: https://doi.org/10.7554/eLife.44702.011

The following figure supplements are available for figure 4:

**Figure supplement 1.** De novo fatty acid synthesis by oligodendrocytes supports structural stability of myelinated axons in the optic nerve.

DOI: https://doi.org/10.7554/eLife.44702.012

**Figure supplement 2.** Lipid profiles of spinal cord myelin of adult control and FASN mutant mice.

DOI: https://doi.org/10.7554/eLife.44702.013

of energy by oxidation (*Lodhi and Semenkovich, 2014*), through palmitoylation of proteins (*Resh, 2016*; *Salaun et al., 2010*), and by modulating transcriptional networks (*Ahmadian et al., 2013*). In this context, we have previously identified activation of PPARγ as a mediator of FASN in triggering onset of PNS myelination (*Montani et al., 2018*). In view of these findings, molecular alterations due to lack of FASN in the OL lineage appeared likely. Given that we had not observed significant alterations in OL differentiation up to the CC1+ stage, such potential changes are predicted to reflect functions of OL-derived FASN in the transition of CC1+ OLs to the final stages of maturation up to myelination completion. We addressed this issue by comparing the optic nerve transcriptome of P14 mutant and control mice (*Figure 5—source data 1*). In line with our morphological findings, Metacore enrichment analysis revealed repression of GeneOntology Processes associated with development of the nervous system, OL differentiation, and myelination (*Figure 5a,b*). A more detailed Metacore Pathway analysis highlighted downregulation of known regulatory axes of OL maturation and myelination, including cytoskeleton proteins, cell-cell and ECM-cell interactions, as well as epigenetic and transcriptional regulation (*Figure 5a,b*). To explore these dysregulations further, we compared them to OLs single cell transcriptomic data (*Marques et al., 2016*). This comparison revealed that mutant optic nerves displayed increased expression of a subset of transcripts enriched in mature OLs (MOL 1–6 in ref. *Marques et al., 2016*) such as Klk6, Sepp1, and Glul. In parallel, reduced expression of several transcripts enriched in myelin-forming OLs (MFOL1/2 in ref. *Marques et al., 2016*) including Ctps, Sirt2, Gpr17, Olig1, Mag and Mobp was identified. Following up on these findings by a candidate approach, we detected also reduced expression of transcripts encoding well known myelin-associated proteins such as PLP1, MBP and CNP, and of major regulators of OLs including Sox10, Sox8, and Myrf (*Bujalka et al., 2013*; *Hornig et al., 2013*; *Li and Richardson, 2016*; *Turnescu et al., 2018*) (*Figure 5c, Figure 5—figure supplement 1*). Combined with our previous findings, the dysregulated molecular fingerprint revealed by these data is consistent with a functional role for FASN in late OL maturation and myelination. However, the impaired myelination in mutant mice may also contribute to the transcriptome differences compared to controls.

## Increased dietary intake of lipids can partially rescue lack of fatty acid synthesis by oligodendrocytes

Increasing dietary intake of lipids was unable to rescue lack of endogenous FA synthesis by Schwann cells in PNS myelination, instead being surprisingly detrimental (*Montani et al., 2018*). To investigate whether such a treatment could rescue myelination if OLs are deficient in FA synthesis, we fed mutant and control mice either a 60% high fat diet (HFD; % of caloric intake from fat, enriched in palmitic acid; complete FA content in Materials and methods), previously shown to be capable of rescuing diminished lipogenesis in astrocytes in vivo (*Camargo et al., 2012*; *Camargo et al., 2017*), or a standard diet (STD; complete FA content in Materials and methods). Pregnant females and subsequently born pups were fed with the different diets from E14 up to P40, the time of analysis (*Figure 6a*). We found that the HFD regimen lead to improved myelin radial growth in the spinal cord of mutant mice compared to mutant mice fed with STD (*Figure 6b,c and d*). There were no

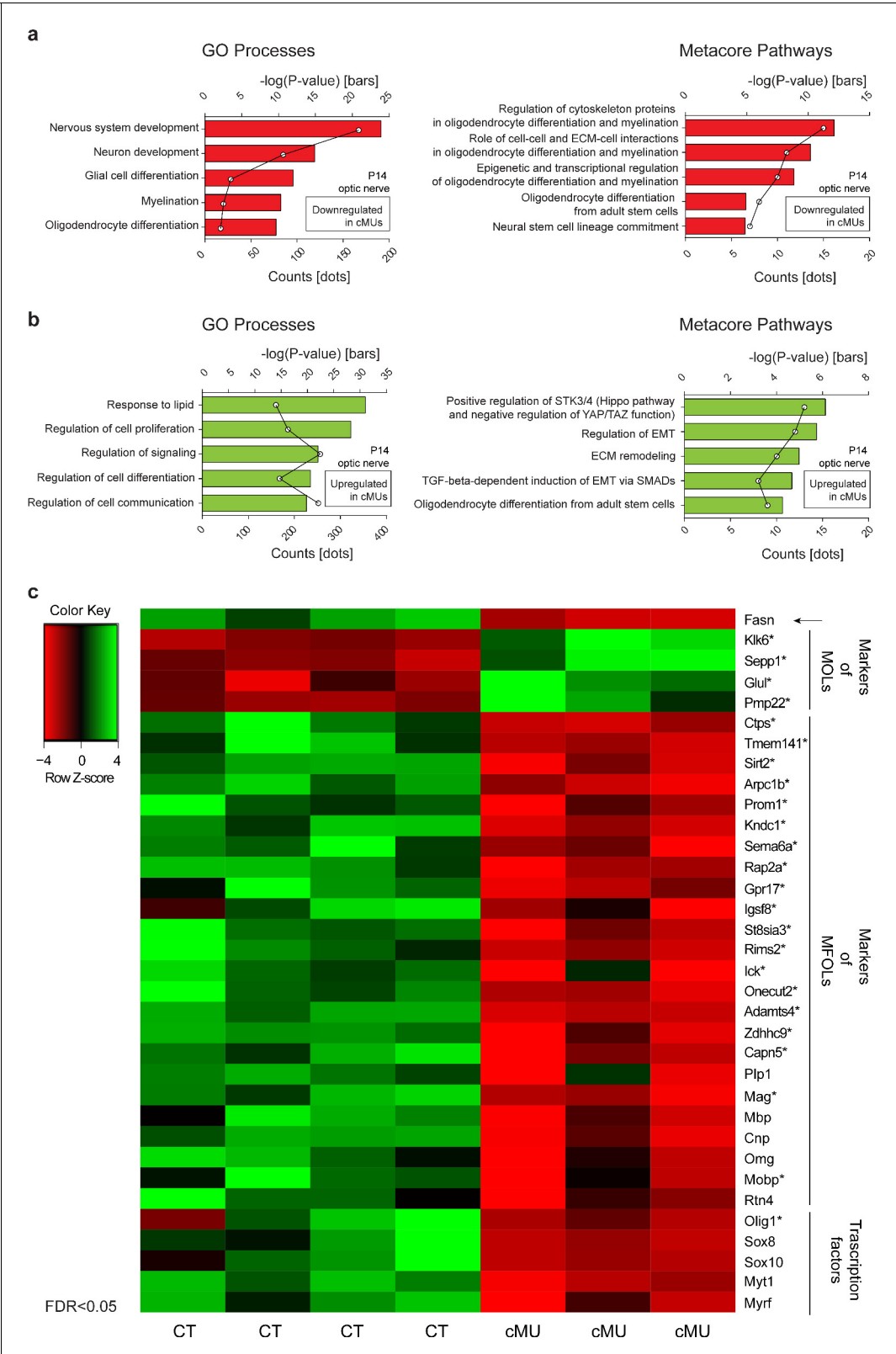

**Figure 5.** Transcriptome analysis of optic nerves of *Fasn* mutant mice reveal oligodendrocyte defects in late stages of maturation, including myelination. (**a, b**) GeneOntology analysis depicting biological processes affected by down-regulated (**a**) and up-regulated (**b**) transcripts in P14 optic nerves of mutant (cMU) mice. Metacore Pathway analysis showing pathways affected by downregulated (**a**) and upregulated (**b**) transcripts in P14 optic nerves of cMU mice. Data points represent the count of regulated transcript for each category and pathway. (**c**) Heat map of RNA-seq data of CT and

*Figure 5 continued on next page*

Figure 5 continued

cMU optic nerves at P14, depicting Fasn and selected down- and up-regulated modulators of OL differentiation, maturation and myelination (FDR < 0.05). Markers identified by comparison with OLs single cell sequencing data (*Marques et al., 2016*) are marked with an asterisk. n = 3 mice for cMU, n = 4 mice for CT. CT = control, cMU = conditional mutant, MFOLs = myelin-forming oligodendrocytes, MOLs = mature oligodendrocytes (*Marques et al., 2016*).

DOI: https://doi.org/10.7554/eLife.44702.014

The following source data and figure supplement are available for figure 5:

**Source data 1.** Expressed transcripts in optic nerves from control and cMU mice at P14.

DOI: https://doi.org/10.7554/eLife.44702.016

**Figure supplement 1.** FASN deficiency is correlated with molecular signs of defective oligodendrocyte maturation.

DOI: https://doi.org/10.7554/eLife.44702.015

significant effects of HFD on radial myelin growth in control mice (*Figure 6b,c and d*). Myelinated fibers in the spinal cord of mutant mice treated with HFD were still hypomyelinated compared to those of control mice under HFD or STD, but less so than those of mutant mice under STD (*Figure 6b,c and d*). We conclude that lack of endogenous FA synthesis in OLs during development can be partially compensated by increasing dietary lipid intake.

## Depletion of fatty acid synthase in adult oligodendrocyte progenitors to study CNS remyelination

Following demyelination, new myelin is formed by aOPCs (*Crawford et al., 2016*; *Franklin and Ffrench-Constant, 2017*). To address the significance of FA synthesis in this process, we conditionally ablated the *Fasn* gene specifically in aOPCs of transgenic mice using tamoxifen-activatable Cre recombinase under the control of *Pdgfrα* gene regulatory elements (*Rivers et al., 2008*). Recombination also activated YFP expression via the Cre-dependent reporter *Rosa26-loxPstoploxP-YFP* allele (*Srinivas et al., 2001*) that was included in both, induced mutants (i-cMU) and induced control (CT) mice. This experimental design allowed us to follow the fate of recombined cells (*Figure 7a*; *Figure 7—figure supplement 1a,b*). We then addressed how loss of FASN in aOPCs affects remyelination of a demyelinated focal lesion in the spinal cord, induced by injection of the gliotoxin lysolecithin (experimental setting schematized in *Figure 7b*). Such lesions undergo a stereotyped timed program of remyelination that is characterized by high proliferation of recruited aOPCs, followed by their differentiation into CC1+ OLs, OL maturation, and onset of remyelination within the first 10–14 days post lesion (dpl), which is largely completed by 21 dpl (*Fancy et al., 2011*; *Zawadzka et al., 2010*) (*Figure 7b*). Immunohistochemistry revealed high FASN expression in differentiated CC1+ aOPC-derived OLs in control mice at the lesion site 7 dpl (*Figure 7c,d*). As expected, this expression was lost in mutant mice (*Figure 7c,d*).

## Fatty acid synthesis is required to sustain aOPC-derived oligodendrocytes during remyelination

To explore the role of FA synthesis in OLs during remyelination, we next analyzed FASN expression in aOPCs and the effects of its depletion on their differentiation following demyelination. Comparable to our developmental studies, we found only marginal expression (if any) of FASN in aOPCs located in acute lesions at 7 dpl (*Figure 8—figure supplement 1a*). Statistical analysis of immunohistochemical data revealed no significant difference between induced mutant and control mice in numbers of recombined aOPCs (YFP+ PDGFRα+ Olig2+) when data were analyzed over time at 7, 10, 14 and 21 dpl (*Figure 8—figure supplement 1b,c*). However, if mutants and controls were statistically compared at the individual time points, a modest but significant reduction in recombined aOPCs was apparent at 14 dpl and 21 dpl (unpaired two-tailed two sample Student's t-test; i-cMU vs. CT, at 14 dpl: p=0.0164, $t = 3.027$; at 21 dpl: p=0.041, $t = 2.434$). We conclude that FASN is mostly dispensable for building up and maintaining the pool of aOPCs at the lesion site, given that in our lesion model this pool is largely established at 10 dpl (*Figure 8—figure supplement 1b,c*). In line with this interpretation, and similar to the developmental situation, we also found that FASN is not essential for early proliferation of OL lineage cells (Olig2+) as tested by Ki67 immunohistochemistry at 7 dpl (*Figure 8—figure supplement 2a,b*).

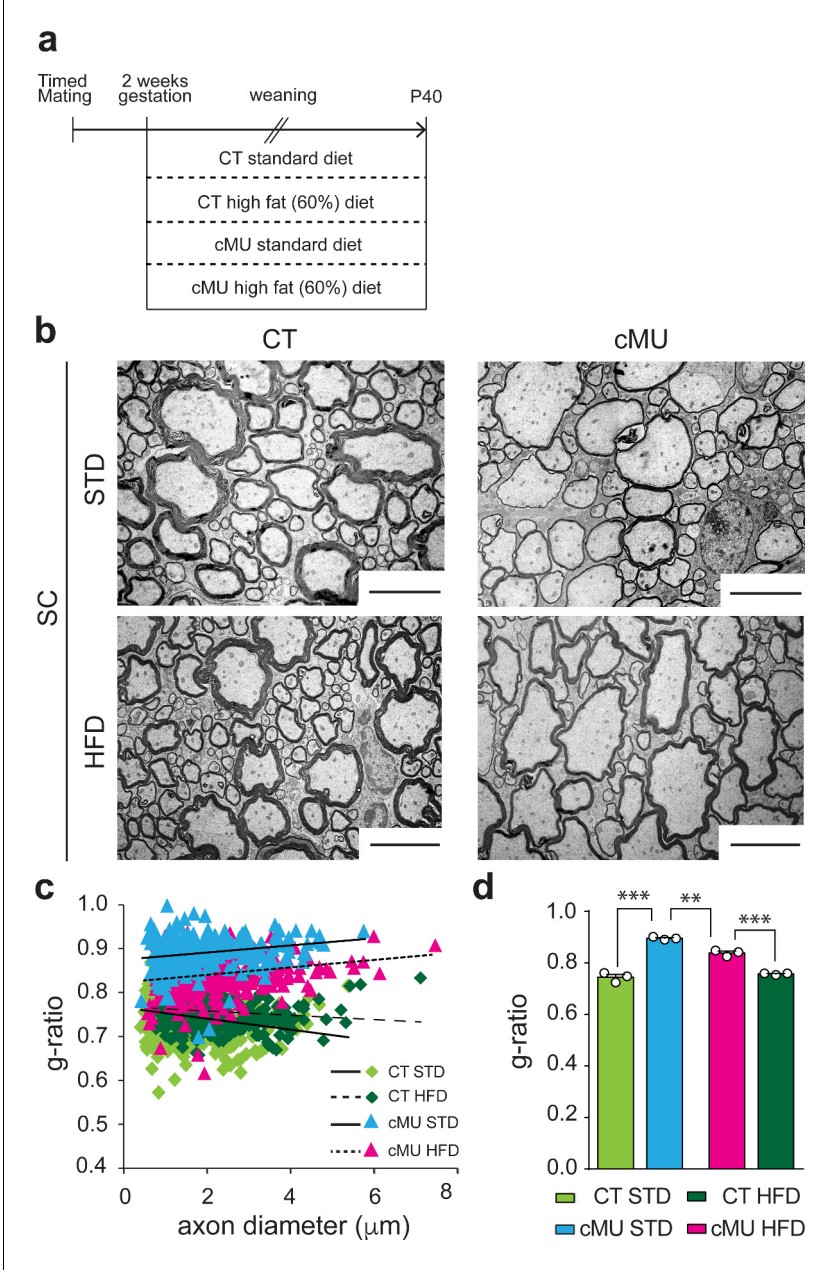

**Figure 6.** Increasing dietary lipid intake can partially compensate defects in radial myelin growth caused by lack of endogenous fatty acid synthesis in oligodendrocytes. (a) Scheme of the experimental design. (b) Exemplary EM micrographs of the ventral funiculus of the lumbar spinal cord from mutant (cMU) and control (CT) mice that were fed a standard (STD) or a high-fat (HFD) diet. Scale bars: 5 μm. (c) Linear correlation of g-ratio versus axon diameter. At least 84 myelinated axons analyzed per mouse, in random fields selected in the same anatomical area, from $n = 3$ mice for each, CT and cMU, STD and HFD. (d) Overall hypomyelination in cMU compared to CT fed STD at P40, and partial recovery of radial myelination in cMU fed with HFD, as shown by g-ratio analysis. Data points represent $n = 3$ mice for each, CT and cMU, STD and HFD (one-way Anova; Treatment: p<0.0001, $F_{3,8}$ = 113.2; with Sidak's multiple comparisons test; cMU STD vs. CT STD: p<0.0001, $t$ = 15.96; cMU HFD vs. CT HFD: p=0.0002, $t$ = 8.597; CT HFD vs CT STD: p=0.7703, $t$ = 1.339; MU HFD vs. MU STD: p=0.0019, $t$ = 6.025), ***p<0.001, **p<0.01. Bars represent mean ±SEM. STD = standard diet, HFD = high fat diet, CT = control, cMU = conditional mutant.

DOI: https://doi.org/10.7554/eLife.44702.017

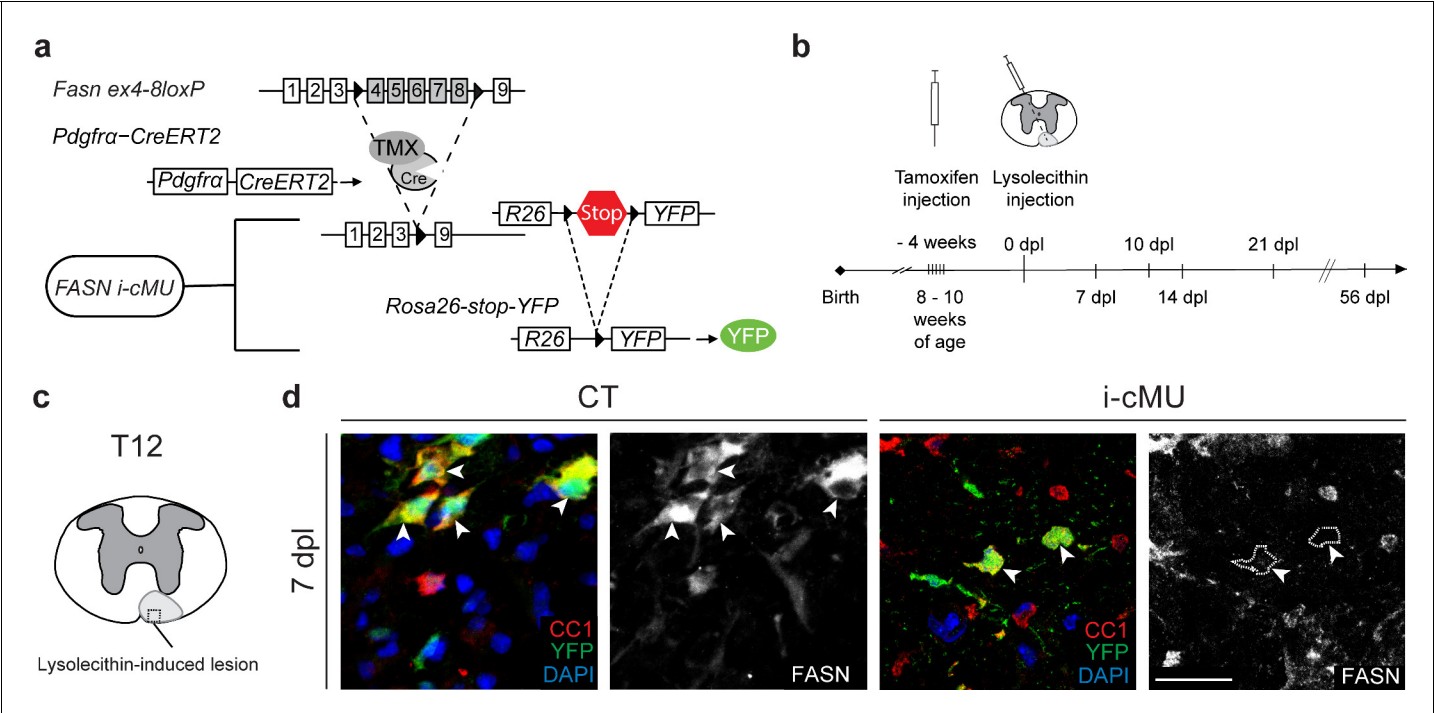

**Figure 7.** Generation of tamoxifen-inducible *PdgfraCreERT2:FASN-floxed* conditionally mutant mice. (**a**) Tamoxifen-mediated nuclear translocation of *Pdgfra*-driven CreERT2 induces conditional *Fasn* allele inactivation (i-cMU) and expression of yellow fluorescent protein (YFP). (**b**) Timeline of de-/re-myelination experiments: Focal demyelination of the ventral spinal cord white matter was induced by injection of lysolecithin 4 weeks post-tamoxifen administration. The tissue was analysed 7, 10, 14, 21 and 56 days post-lysolecithin injection (dpl). (**c**) Schematic of lysolecithin-induced demyelinated focal lesion in the ventral funiculus of the thoracic spinal cord (level T12). Square insert indicates the area where immunostaining images were acquired. (**d**) Representative immunostainings of lesioned areas in cross-sectioned thoracic spinal cords from CT and i-cMU mice 7 dpl, *n* = 3 mice analyzed for each, CT and i-cMU. Note the prominent FASN expression in the cytoplasm of recombined aOPC-derived differentiated OLs (CC1+ YFP+ FASN+, examples indicated by arrowheads) in CT, but not in i-cMU (CC1+ YFP+ FASN-, examples indicated by dotted lines) lesions. Nuclear marker: DAPI. Scale bar: 20 μm, applies to entire panel. CT = control, i-cMU = inducible conditional mutant, dpl = days post-lysolecithin injection.
DOI: https://doi.org/10.7554/eLife.44702.018

The following figure supplement is available for figure 7:

**Figure supplement 1.** Recombination efficiency in spinal cords of *PdgfraCreERT2:Rosa26-loxPstoploxP-YFP* control animals induced with tamoxifen.
DOI: https://doi.org/10.7554/eLife.44702.019

Next, we examined whether FASN is required during differentiation and maturation of aOPCs into myelinating OLs following demyelination. We found no difference in the number of recombined total (Olig2+ YFP+) and differentiated (CC1+ Olig2+ YFP+) aOPC-derived OLs in mutant and control mice during early differentiation (7 dpl and 10 dpl) (*Figure 8a,b and c*). However, the number of both recombined total (Olig2+ YFP+) and differentiated (CC1+ Olig2+ YFP+) OLs was substantially lower in mutants compared to controls when remyelination was ongoing (14 dpl), with no major recovery at 21 dpl (Olig2+ YFP+ cells per area (mm$^2$) at 14 dpl: 98.4 ± 13.71 in mutants vs. 294.60 ± 28.05 in controls; at 21 dpl: 108.10 ± 11.93 in mutants vs. 237.70 ± 22.55 in controls; CC1+ Olig2+ YFP+ cells per area (mm$^2$) at 14 dpl: 45.18 ± 6.71 in mutants vs. 193.20 ± 11.73 in controls; at 21 dpl: 74.60 ± 5.00 in mutants vs. 183.00 ± 18.47 in controls) (*Figure 8a,b and c*). This impairment was present already at the onset of remyelination (12 dpl; *Figure 8—figure supplement 3a,b and c*), suggesting that OL death might be involved. Cleaved caspase 3 (cC3) immunostaining at 11 dpl did not detect a significant increase in the percentage of apoptotic differentiated recombined OLs (cC3+ CC1+ YFP+) in lesions of mutants compared to controls (*Figure 8—figure supplement 3d,e*). However, we found such cells at very low frequency, contributing to a rather high variability between animals and limiting the power of our analysis and its interpretation. Thus, we assume that our inability to find an expected significant increase in cC3-associated OL cell death in

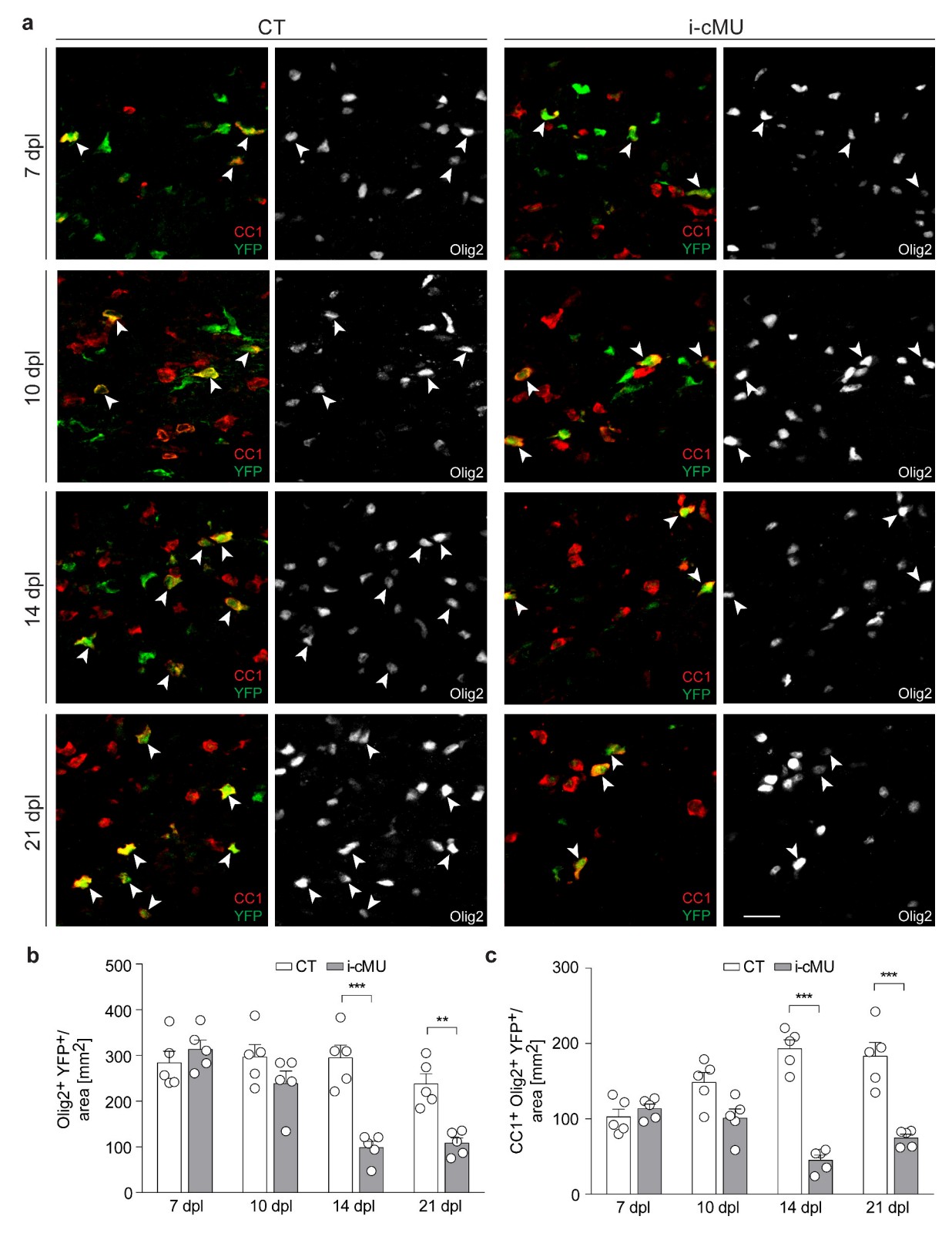

**Figure 8.** De novo fatty acid synthesis is required for maintaining the adult oligodendrocyte progenitor-derived oligodendrocyte population during remyelination. (a) Representative immunostaining of lesions of control (CT) and inducible conditional mutant (i-cMU) mice on cross sections of ventral white matter of the spinal cord at 7, 10, 14 and 21 days post-lysolecithin injection (dpl), $n$ = 5 mice for each, CT and i-cMU. Images depict recombined aOPC-derived differentiated OLs (CC1+ Olig2+ YFP+, examples indicated by arrowheads). Scale bar: 20 µm, applies to entire panel. (b, c)

*Figure 8 continued on next page*

*Figure 8 continued*

Corresponding graphs with quantification of (**b**) recombined OLs (Olig2+ YFP+) and (**c**) recombined aOPC-derived differentiated OLs (CC1+ Olig2 + YFP+) in lesion areas of ventral spinal cord white matter of CT and i-cMU mice at 7, 10, 14 and 21 dpl. Data were normalized per area. Data points represent $n$ = 5 mice for each, CT and i-cMU. Lesion areas of at least 4 sections quantified per animal (2-way Anova; (**b**) Genotype: p<0.0001, $F_{1,32}$= 30.28; (**c**) Genotype: p<0.0001, $F_{1,32}$= 84.52; with Tukey's multiple comparisons test; (**b**) 7 dpl: i-cMU vs. CT, p=0.9829; 10 dpl: i-cMU vs. CT, p=0.6132; 14 dpl: i-cMU vs. CT, p<0.0001; 21 dpl: i-cMU vs. CT, p=0.0071; (**c**) 7 dpl: i-cMU vs. CT, p=0.9971; 10 dpl: i-cMU vs. CT, p=0.0900; 14: i-cMU vs. CT, p<0.0001; 21 dpl: i-cMU vs. CT, p<0.0001), ***p<0.001, **p<0.01. Bars represent mean ± SEM. CT = control, i-cMU = inducible conditional mutant, dpl = days post-lysolecithin injection.
DOI: https://doi.org/10.7554/eLife.44702.020

The following figure supplements are available for figure 8:

**Figure supplement 1.** FASN expression in adult oligodendrocyte progenitors is marginal and largely dispensable for their response following demyelination.
DOI: https://doi.org/10.7554/eLife.44702.021

**Figure supplement 2.** FASN expression is dispensable for the proliferation of oligodendrocytes lineage cells in remyelination.
DOI: https://doi.org/10.7554/eLife.44702.022

**Figure supplement 3.** FASN is critical to sustain adult oligodendrocyte progenitor-derived oligodendrocytes during remyelination.
DOI: https://doi.org/10.7554/eLife.44702.023

mutants is likely due to either very fast removal of dying cells or cell death not marked by cC3-positivity, possibilities that we were unable to explore further due to technical limitations.

Overall our data indicate a crucial role for FA synthesis in sustaining aOPC-derived OLs during remyelination. However, we cannot rule out that also the mild reduction in the number of aOPCs in mutants at 14 dpl and 21 dpl makes some contribution to the observed phenotype. In this context, we favor the interpretation that the observed decrease in CC1+ recombined OLs in mutant animals during remyelination may promote a mild reaction of aOPCs, potentially with the goal of some compensation. If the recombined cells cannot be sustained and die while differentiating from aOPCs towards myelinating oligodendrocytes, this may cause a slight reduction of the remaining numbers of recombined aOPCs. Alternatively, aOPCs may have intrinsic differences in their dependency on FASN compared to developmental OPCs with regard to survival. We consider this less likely given the low expression of FASN by developmental OPCs and aOPCs.

## Fatty acid synthesis is essential for efficient CNS remyelination

Based on our findings, we next investigated whether the observed drop in differentiated OLs causes a failure of accurate remyelination in mutants. Thus, we analyzed the ultrastructural morphology of remyelinating spinal cord lesions by comparative EM as remyelinated axons can be distinguished by their relatively thin myelin compared to developmentally achieved myelination (*Franklin and Ffrench-Constant, 2017*). Consistent with the reduced number of differentiated CC1+ OLs present during the active remyelination phase at 14 dpl, we found that mutant mice displayed more axons not encased by myelin (*Figure 9a*, false colored) compared to control mice (*Figure 9a*). Quantification confirmed these observations (35.63 ± 3.42% demyelinated axons in mutants vs. 22.68 ± 1.58% in controls) (*Figure 9b*). Also at 21 dpl, only 8.24 ± 0.70% of axons remained demyelinated in control mice, compared to 18.00 ± 1.91% in mutants. At this time point, focal lesions approach their full remyelination potential (*Crawford et al., 2016*; *Franklin and Ffrench-Constant, 2017*). In agreement, further analysis of a later time point (56 dpl) revealed only a slight additional decrease in the percentage of demyelinated fibers in control mice to 5.58 ± 1.02% (*Figure 9b*). Similarly, this value decreased also in mutants to 12.04 ± 1.11% (*Figure 9b*). However, in direct comparison of controls and mutants at 56 dpl, mutants still retained a larger percentage of demyelinated axons (12.04 ± 1.11% in mutants vs. 5.58 ± 1.02% in controls) (*Figure 9b*). Remyelinated profiles of both control and mutant mice showed characteristically thin myelin (*Figure 9a*) (*Franklin and Ffrench-Constant, 2017*), with no significant difference detectable in the thickness of myelin by average g-ratio analysis (*Figure 9c*) and by linear correlation of g-ratio versus axon diameter (*Figure 9d*) at the sampled time points (14, 21 and 56 dpl). We conclude that FASN and consequently FA synthesis in OLs is essential for efficient remyelination in adult mice.

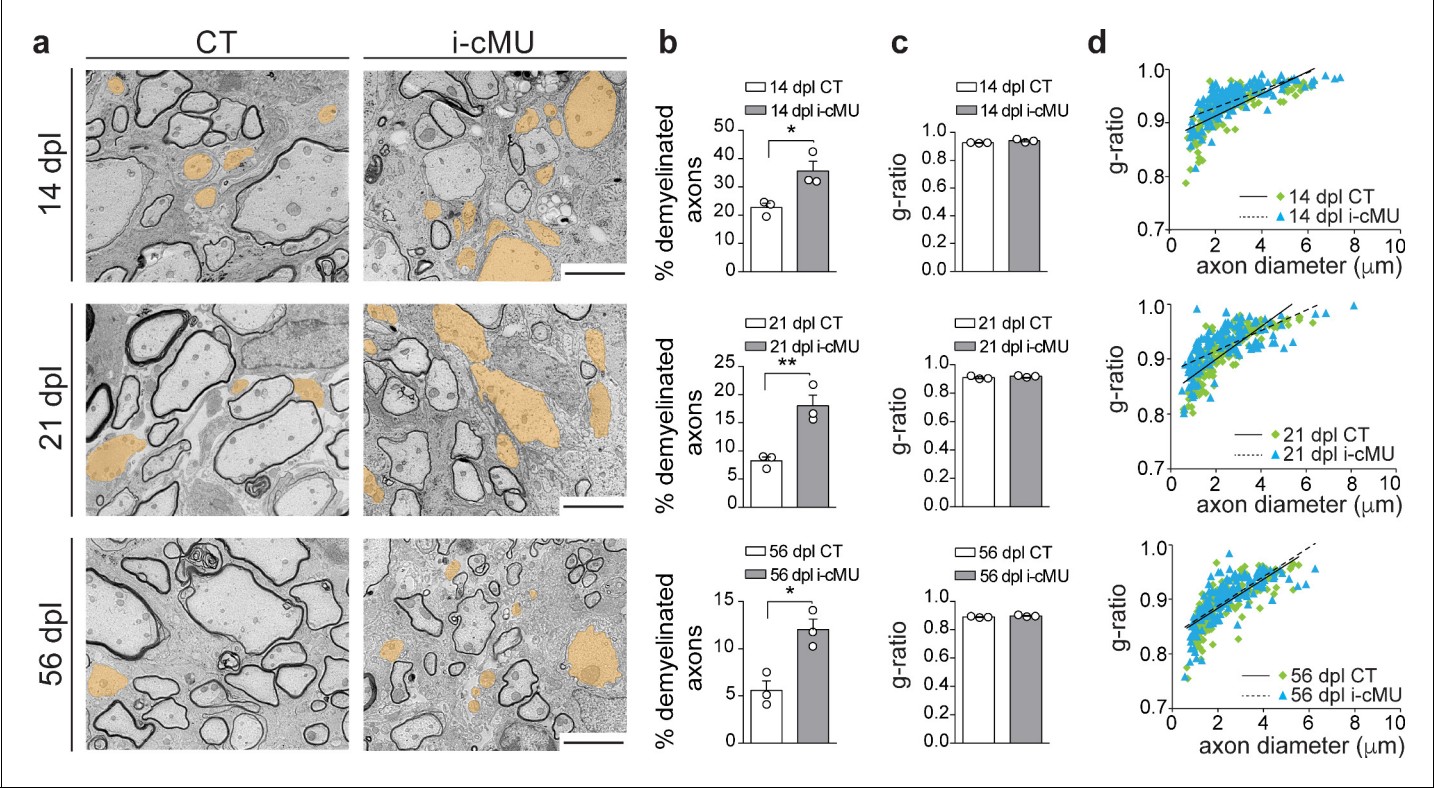

**Figure 9.** De novo fatty acid synthesis is essential to achieve efficient remyelination in the spinal cord. (a) Representative EM images of lesions in the spinal cord of control (CT) and inducible conditional mutant (i-cMU) mice during remyelination at 14, 21 and 56 days post-lysolecithin injection (dpl). i-cMUs show persistently increased numbers of demyelinated axons (examples false colored in orange) compared to CTs up to 56 dpl. Scale bars: 3 μm. (b) Corresponding graphs with quantification of percentage of demyelinated axons within the remyelinated lesion area at 14, 21 and 56 dpl. Data points represent $n = 3$ mice for each, CT and i-cMU. The entire lesion area was quantified, at least 550 axons counted per animal (unpaired two-tailed two sample Student's t-test; 14 dpl: i-cMU vs. CT, p=0.0263, $t = 3.439$; 21 dpl: i-cMU vs. CT, p=0.0086, $t = 4.803$; 56 dpl: i-cMU vs. CT, p=0.0127, $t = 4.290$), *p<0.05, **p<0.01. (c) g-ratio analysis of remyelinated fibers at 14, 21 and 56 dpl in i-cMU compared to CTs. Data points represent $n = 3$ mice for each, CT and i-cMU (unpaired two-tailed two sample Student's t-test; 14 dpl: i-cMU vs. CT, p=0.1146, $t = 2.011$; 21 dpl: i-cMU vs. CT, p=0.3146, $t = 1.149$; 56 dpl: i-cMU vs. CT, p=0.1481, $t = 1.789$). (d) Linear correlation of g-ratio versus axon diameter in remyelinating lesions of i-cMU compared to CT mice, at 14, 21 and 56 dpl. At least 50 myelinated axons per mouse analyzed, $n = 3$ mice for each, CT and i-cMU. Bars represent mean ± SEM. CT = control, i-cMU = inducible conditional mutant, dpl = days post-lysolecithin injection.

DOI: https://doi.org/10.7554/eLife.44702.024

## Discussion

Oligodendrocytes require rapid access to large quantities of lipids over a short period of time to achieve accurate production of the considerable membrane mass required to myelinate multiple axons (*Chrast et al., 2011*; *Nave and Werner, 2014*; *Schmitt et al., 2015*). To which extent this demand is supported by endogenous synthesis in OLs and/or lipid uptake is not fully understood (*Chrast et al., 2011*; *Schmitt et al., 2015*). In support of a critical role for OL-endogenous lipid bio-synthesis, the mTORC1/SREBP signaling axis was suggested to be critical for efficient CNS myelina-tion (*Camargo et al., 2017*; *Lebrun-Julien et al., 2014*). Furthermore, OLs depend on cholesterol synthesis for efficient radial growth of myelin (*Saher et al., 2005*). Postnatally, newly added choles-terol – the major sterol present in myelin – can only be synthesized within the CNS, while the other main myelin lipids share FAs as building blocks that might be either locally synthesized or taken up from the bloodstream (*Currie et al., 2013*; *Harayama and Riezman, 2018*). Here, we have addressed the need for FA synthesis in OLs by depleting FASN in transgenic mice. This multifunc-tional enzyme is responsible for the synthesis of palmitate, the substrate for the production of more complex FAs (*Currie et al., 2013*; *Harayama and Riezman, 2018*). We show that lack of FASN in

OLs results in deficits in CNS myelination, including a decreased capacity for myelin to grow. Thus, OLs strictly depend on de novo endogenous FA synthesis to achieve normal myelination.

Another source for non-essential FAs, and the only source available for essential ones, is the uptake from the FA pool present in the circulation (*Harayama and Riezman, 2018*). We have previously demonstrated that increasing dietary lipids by administering a HFD cannot ameliorate the dysmyelination resulting from lack of FASN in myelinating Schwann cells of the PNS (*Montani et al., 2018*). Thus, we were somewhat surprised to find that the same HFD experimental regimen was able to achieve partial amelioration of the CNS hypomyelination caused by lack of FASN in OLs. Similarly, it has been shown that the HFD used in our study could partially substitute for the lack of SCAP in another CNS cell population, the astrocytes (*Camargo et al., 2012*; *Camargo et al., 2017*). The causative mechanisms for these differences between the PNS vs. CNS remain to be determined. Future studies will need to take into account that the two tissues appear to have evolved distinct compensatory mechanisms to deal with this situation. Specifically, adipocytes that are present in the layers surrounding peripheral nerves (i.e. epineurial adipocytes) can carry out lipolysis, potentially to provide supporting lipids when endogenous synthesis in Schwann cells fails (*Montani et al., 2018*) or might become limiting. In contrast, the CNS contains astrocytes which can provide horizontal lipid flux to OLs (*Camargo et al., 2017*). However, the specifics of qualitative (i.e. which lipid species are involved) and quantitative contributions, both spatially and temporally, are challenging open research questions. In addition, the relevance of such mechanisms to the maintenance of homeostatic physiology and in disease situations remain important topics to be investigated.

Given the relevance of efficient remyelination as a drug target in diseases such as multiple sclerosis (*Cole et al., 2017*; *Franklin and Ffrench-Constant, 2017*) and the currently clinically explored potential of biotin – an important regulator of FA synthesis – in multiple sclerosis treatment (*Sedel et al., 2016*), we were motivated to evaluate the significance of FA synthesis by OLs also in remyelination. To realize this goal, we used the focal lysolecithin-induced demyelinating-remyelinating lesion model in the spinal cord of transgenic mice after depleting FASN specifically in aOPCs. We show that de novo synthesis of FAs is crucial to sustain remyelinating aOPC-derived OLs and is ultimately required for successful remyelination. Specifically, we found that after being able to differentiate correctly, the number of aOPCs-derived OLs was substantially reduced in mutants compared to controls at onset of remyelination, persisting at later time points. Thus, endogenous FA synthesis appears to be pivotal to maintain survival of newly differentiated and mature aOPCs-derived OLs when myelination starts. At the myelin level, depletion of FASN in aOPCs and their progenies was associated with reduced numbers of remyelinated axons in mutants compared to controls from 14 dpl (first time point analyzed) up to 56 dpl (latest time point analyzed). This rather strong phenotype was not expected, since we anticipated that partial disruption of the blood-brain (-spinal cord) barrier in the lesioned CNS may lead to facilitated access of OLs to supplementary FA sources, that is from the pool of FAs in the blood, either directly or from increased horizontal flux from astrocytes (*Camargo et al., 2017*; *Saher et al., 2012*). Possibly, saturation of molecules mediating FA-uptake and/or unknown regulatory mechanisms might restrict the FA uptake capacity of OLs not allowing full compensation for the lack of endogenous synthesis. Such mechanisms may also be differentially regulated in lesioned versus intact tissue. Future studies in this context should be aimed at achieving a detailed understanding of the molecular mechanisms that regulate synthesis and trafficking of FAs in the normal and diseased CNS, with a particular emphasis on OLs, astrocytes and blood vessels, including endothelial cells and pericytes. Such knowledge is expected to broaden our understanding of the fundamental mechanisms that regulate developmental myelination, pathological states in diseases, and remyelination after injury (*Chrast et al., 2011*; *Schmitt et al., 2015*).

## Materials and methods

**Key resources table**

| Reagent type (species) or resource | Designation | Source or reference | Identifiers | Additional information |
|---|---|---|---|---|

*Continued on next page*

*Continued*

| Reagent type (species) or resource | Designation | Source or reference | Identifiers | Additional information |
|---|---|---|---|---|
| Genetic reagent (*M. musculus*) | *Fasnlox/lox* | PMID:16054078 | MGI:3765070 | Dr. Clay F. Semenkovich, Washington University, St. Louis USA |
| Genetic reagent (*M. musculus*) | *PdgfraCreERT2* | PMID:18849983 | MGI:3832569 | Dr. William D. Richardson, University College London, London UK |
| Genetic reagent (*M. musculus*) | *Olig2Cre* | Jackson Laboratory | Stock#:011103; MGI:3810299 | PMID:18691547 |
| Genetic reagent (*M. musculus*) | *Rosa26-lox PstoploxP-YFP* | Jackson Laboratory | Stock#:006148; MGI:3621481 | PMID:11299042 |
| Sequence-based reagent | Genotyping primer: Fasn forward | PMID:29434029 | | 5'-GGATAGCTGTGTAGTGTAACCAT-3' |
| Sequence-based reagent | Genotyping primer: Fasn reverse | PMID:29434029 | | 5'-GGTCACCCAGCAGGAAAGGGC- 3' |
| Sequence-based reagent | Genotyping primer: Cre forward | PMID: 29434029 | | 5'-TTCCCGCAGAACCTGAAGATGTTCG-3' |
| Sequence-based reagent | Genotyping primer: Cre reverse | PMID: 29434029 | | 5'-GGGTGTTATAAGCAATCCCCAGAAATG-3' |
| Sequence-based reagent | Genotyping primer: Rosa26-lox PstoploxP-YFP forward | PMID: 28522536 | | 5'-AAAGTCGCTCTGAGTTGTTAT-3' |
| Sequence-based reagent | Genotyping primer: Rosa26-loxPs toploxP-YFP reverse transgenic | PMID: 28522536 | | 5'-GCGAAGAGTTTGTCCTCAACC-3' |
| Sequence-based reagent | Genotyping primer: Rosa26-loxPstoploxP-YFP reverse wildtype | PMID: 28522536 | | 5'-GGAGCGGGAGAAATGGATATG-3' |
| Sequence-based reagent | qRT-PCR primer: SOX10 Forward | This paper | | 5'-CCGACCAGTACCCTCACCT-3' |
| Sequence-based reagent | qRT-PCR primer: SOX10 Reverse | This paper | | 5'- TCAATGAAGGGGCGCTTGT-3' |
| Sequence-based reagent | qRT-PCR primer: MYRF Forward | This paper | | 5'-ATGGAGGTGGTGGACGAGAC-3' |
| Sequence-based reagent | qRT-PCR primer: MYRF Reverse | This paper | | 5'-GGCGTCCTCTTTGCCAATGT-3' |
| Sequence-based reagent | qRT-PCR primer: MBP Forward | This paper | | 5'-ACACGAGAACTACCCATTATGGC-3' |
| Sequence-based reagent | qRT-PCR primer: MBP Reverse | This paper | | 5'-CCAGCTAAATCTGCTGAGGGGA-3' |
| Sequence-based reagent | qRT-PCR primer: Actin Forward | PMID: 28880149 | | 5'-GTCCACACCCGCCACC-3' |

*Continued on next page*

*Continued*

| Reagent type (species) or resource | Designation | Source or reference | Identifiers | Additional information |
|---|---|---|---|---|
| Sequence-based reagent | qRT-PCR primer: Actin Reverse | PMID: 28880149 | | 5'-GGCCTCGTCACCCACATAG-3' |
| Antibody | Rabbit polyclonal anti-cleaved Caspase 3 | Cell signaling Technology, Danvers, MA, USA | Cat# 9661; RRID:AB_2341188 | 1:500 dilution |
| Antibody | Mouse monoclonal anti-CC1 | Merck Millipore, Billerica, MA, USA | Cat; # OP80; RRID:AB_2057371 | Development: 1:200 dilution; Remyelination 1:300 dilution |
| Antibody | rabbit polyclonal anti-FASN | Abcam, UK | Cat# Ab22759; RRID:AB_732316 | 1:200 dilution |
| Antibody | Chicken polyclonal anti-GFP | Abcam, UK | Cat# Ab13970; RRID:AB_300798 | 1:1000 dilution |
| Antibody | mouse monoclonal anti-ki67 | Dako Agilent, Santa Clara, CA, USA | Cat# m7249; clone MIB-5; RRID:AB_2250503 | 1:200 dilution |
| Antibody | rat monoclonal anti-MBP | Serotec/BioRad Laboratories, Hercules, CA, USA | Cat# MCA409S; RRID:AB_325004 | Development: 1:200 dilution; Remyelination 1:300 dilution |
| Antibody | Goat polyclonal anti-Olig2 | R and D Systems, Minneapolis, MN, USA | Cat# AF2418; RRID:AB_2157554 | 1:25 dilution |
| Antibody | mouse monoclonal anti-Olig2 | Merck Millipore, Billerica, MA, USA | Cat# MABN50; clone 211F1.1; RRID:AB_10807410 | 1:1000 dilution |
| Antibody | rabbit polyclonal anti-Olig2 | Merck Millipore, Billerica, MA, USA | Cat# AB9610; RRID:AB_570666 | Development: 1:500 dilution; Remyelination 1:400 dilution |
| Antibody | Rabbit monoclonal anti-PDGFRα | Cell signaling Technology, Danvers, MA, USA | Cat# 3174; RRID:AB_2162345 | 1:500 dilution |
| Commercial assay or kit | Click-iT EdU Assay | Thermo Fisher Scientific, Waltham, MA, USA | Cat# C10337 | |
| Commercial assay or kit | Mouse blocking reagent | Vector Laboratories, Burlingame, CA, USA | Cat# MKB-2213 | |
| Commercial assay or kit | Streptavidin/ Biotin blocking Kit | Vector Laboratories, Burlingame, CA, USA | Cat# SP-2002 | |
| Commercial assay or kit | Qiagen MiniKit (RNeasy Mini Kit) | Qiagen, Hilden, Germany | Cat# 74104 | |
| Commercial assay or kit | TruSeq Stranded mRNA Sample Prep Kit | Illumina, San Diego, CA, USA | Cat# 20020594 | |
| Commercial assay or kit | Maxima RT-Kit | Thermo Fisher Scientific, Waltham, MA, USA | Cat# K1641 | |
| Chemical compound, drug | Lysolecithin | Sigma-Aldrich, Sant Louis, MO, USA | Cat# L4129 | |
| Chemical compound, drug | Tamoxifen | Sigma-Aldrich, Sant Louis, MO, USA | Cat# T5648 | |
| Chemical compound, drug | Lipid standards | Avanti Polar Lipids, Alabaster, AL, USA | | |

*Continued on next page*

*Continued*

| Reagent type (species) or resource | Designation | Source or reference | Identifiers | Additional information |
|---|---|---|---|---|
| Software, algorithm | Lipid Data Analyzer software | PMID: 29058722 | | |
| Software, algorithm | Photoshop CS5 or CS6 | Adobe | | |
| Software, algorithm | FIJI | ImageJ (http://imagej.nih.gov/ij/) | | |
| Software, algorithm | STAR Aligner(v2.5.1b) | PMID: 23104886 | | |
| Software, algorithm | RSEM (v1.2.22) | PMID: 21816040 | | |
| Software, algorithm | EdgeR | PMID: 19910308 | | |
| Software, algorithm | Metacore (vs6.33) | Thomson Reuters | | |
| Other | Standard diet (STD) | KLIBA NAFAG, Provimi KLIBA, Switzerland | Cat# 3437 | |
| Other | High fat diet (HFD) | KLIBA NAFAG, Provimi KLIBA, Switzerland | Cat# 2127 | |

## Contacts for reagents and resource sharing

Requests for further information and reagents may be directed to the corresponding author Ueli Suter (usuter@biol.ethz.ch). *Fasn*^lox/lox mice are available from the Semenkovich laboratory after executing an MTA with Washington University. The *PdgfrαCreERT2* mice are available from the Richardson laboratory after executing an MTA with the University College London. From Jackson Laboratory are available: *Olig2Cre* mice (JAX Stock #011103) and *Rosa26-loxPstoploxP-YFP* mice (JAX Stock #006148).

## FASN conditional knockout mice

For developmental studies, mice (*Mus musculus*) homozygous for the fatty acid synthase (*Fasn*) floxed allele (*Chakravarthy et al., 2005*) (Strain of origin: 129X1/SvJ, subsequently crossed with 129X1/SvJ * C57BL/6 * DBA) were crossed with mice expressing Cre recombinase under the control of the *Olig2* promoter (*Schüller et al., 2008*) (Strain of origin: C57BL/6) to obtain *Cre⁺:Fasn*^lox/lox mutant mice, and *Fasn*^lox/lox or *Fasn*^lox/wt control mice. For experiments assessing recombination frequencies, a *Rosa26-loxPstoploxP-YFP* allele (*Srinivas et al., 2001*) (Strain of origin: C57BL/6) was included.

For remyelination studies, mice homozygous for the *Fasn* floxed allele (*Chakravarthy et al., 2005*) (Strain of origin: 129X1/SvJ, subsequently crossed with 129X1/SvJ * C57BL/6 * DBA) were crossed with reporter mice homozygous for the *Rosa26-loxPstoploxP-YFP* allele and with mice expressing CreERT2 recombinase, activatable upon tamoxifen injection, under the control of the *Pdgfrα* promoter (*PdgfrαCreERT2*) (*Rivers et al., 2008*) (Strain of origin: C57BL/6). Control mice were *PdgfrαCreERT2⁺:Rosa26-loxPstoploxP-YFP:FASN*^wt/wt for immunohistochemical analysis and *Rosa26-loxPstoploxP-YFP:FASN*^lox/lox for electron microscopy analysis. Mutant mice were *PdgfrαCreERT2⁺:Rosa26-loxPstoploxP-YFP:FASN*^lox/lox for all analyses. All mice were injected with tamoxifen. *Fasn* floxed mice were backcrossed for at least three generations to C57BL/6 background, and *Olig2Cre*, *Rosa26-loxPstoploxP-YFP* and *PdgfrαCreERT2* mice were kept on a C57BL/6 background.

Both male and female mice were used throughout all experiments. Mice were group-caged, kept in a 12 hr light/dark cycle, with water and food provided ad libitum. Littermates and age-matched mice were assigned to experimental groups according to age and genotype. No method of randomization was applied. Animals were fed a standard (STD) (Cat# 3437, KLIBA NAFAG, Provimi KLIBA, Switzerland) or a high-fat diet (HFD) (Cat# 2127, KLIBA NAFAG, Provimi KLIBA, Switzerland), as

previously described (*Montani et al., 2018*). Briefly, pregnant females were fed the high fat diet from gestational day 14 until weaning of the pups 3 weeks after birth. Pups were subsequently separated from the mother and kept on a high fat diet until being sacrificed at 40 days of age. Diets content of fatty acids (in percentage) in HFD vs. STD was: C12 0.03 vs. 0.002, C14 0.440 vs. 0.008, C15 0.000 vs. 0.002, C16 8.220 vs. 0.719, C17 0.000 vs. 0.004, C18 4.500 vs. 0.157, C20 0.020 vs. 0.010, C22 0.000 vs. 0.004, C24 0.000 vs. 0.012, C14:1 0.170 vs. 0.001, C16:1c7 1.000 vs. 0.052, C18:1c9 14.120 vs. 1.024, C20:1c9 0.000 vs. 0.013, C22:1c9 0.000 vs. 0.002, C18:2c9c12 4.810 vs. 2.107, C18:3c9c12c15 0.590 vs. 0.212, C20:4(n-6) 0.560 vs. 0.009, C22:5(n-3) 0.000 vs. 0.001. Cholesterol was present in the same percentage (0.030) in both HFD and STD.

Genotypes were determined by PCR on genomic DNA (*Fasn* primers: 5'-Forward GGATAGCTG TGTAGTGTAACCAT-3', Reverse 5'-GGTCACCCAGCAGGAAAGGGC-3'; *Cre* primers: Forward 5'-TTCCCGCAGAACCTGAAGATGTTCG-3', Reverse 5'-GGGTGTTATAAGCAATCCCCAGAAATG-3', *Rosa26-loxPstoploxP-YFP* primers: Forward 5'-AAAGTCGCTCTGAGTTGTTAT-3', Reverse transgenic 5'-GCGAAGAGT TTGTCCTCAACC-3', Reverse wild type 5'-GGAGCGGGAGAAATGGATATG-3'). All animal experiments were performed with the approval and in strict accordance with the guidelines of the Zurich Cantonal Veterinary Office (ref ZH161/2014, ZH090/2017, ZH03/2012, ZH264/2014, ZH207/2017).

## Tamoxifen administration

CreERT2-mediated recombination in 8 to 10 week-old mice was induced by a single daily intraperitoneal injection of 2 mg tamoxifen (Cat# T5648, Sigma-Aldrich, Sant Louis, MO, USA; 20 mg/ml stock solution in sunflower oil containing 10% ethanol) over 5 consecutive days.

## Focal spinal cord demyelinating lesions

Four weeks after tamoxifen induction, demyelinated lesions were induced in the ventral funiculus of the thoracic spinal cord at intervertebral level T12/T13 as previously described (*Fancy et al., 2011*). Mice received a preoperative subcutaneous injection of 0.1 mg/kg (of body weight) of buprenorphine (Temgesic solution injectable 0.3 mg/ml, Indivior, Switzerland) in Ringer solution (Braun, Switzerland), and were anesthetized by isoflurane inhalation. A dorsal laminectomy was performed at the T12/T13 level, and the dura pierced with an acupuncture needle. 1 µl of 1% lysolecithin (Cat# L4129, Sigma-aldrich, Sant Louis, MO, USA) in 1x PBS, pH 7.4, was injected hemilaterally in the ventral funiculus at a rate of 0.5 µl per minute, with a glass needle coupled to a Hamilton syringe, via a three-way micromanipulator (Narishige, Japan). The overlaying musculature was sutured, and wound clips were applied to close the skin above. Animals were allowed to recover on a heating pad at 37° C. Animals received twice daily a subcutaneous injection of 0.1 mg/kg (of body weight) of buprenorphine (Temgesic solution injectable 0.3 mg/ml, Indivior, Switzerland) for two days following the injury. For additional analgesic treatment, animals were supplied with *ad libitum* 0.01 mg/ml buprenorphine (Temgesic solution injectable 0.3 mg/ml, Indivior, Switzerland) in drinking water for three days following injury.

## TEM and SEM microscopy

Mice were anaesthetized by terminal intraperitoneal injection of pentobarbital, 10% in saline solution (0.9% NaCl) (Eskonarkon, Streuli Pharma, Switzerland). Mice were intracardially perfused with 0.1M phosphate buffer (PB) pH 7.4, followed by a 2.5% glutaraldehyde/4% paraformaldehyde solution in 0.1M PB. After dehydration through an acetone series, tissues were post-fixed in 2% osmium tetroxide overnight, and embedded in Spurr's resin (Electron Microscopy Sciences, Hatfield, PA, USA). Ultrathin sections were cut on a Leica UC-7 (Leica microsystems, Germany) or a Reichert-Jung Ultra cut E ultramicrotome (Leica microsystems, Germany, 65 nm-thick for transmission electron microscopy (TEM) or 99 nm-thick for scanning electron microscopy (TEM)). Sections transferred onto copper grids with a carbon film (Electron Microscopy Sciences, Hatfield, PA, USA) for TEM or ITO coverslips (Optic Balzers, Germany) for SEM, were counterstained with 2% uranyl acetate and 1% lead citrate. Images were acquired with a Morgagni 268 (FEI) for TEM, for analysis of developmental myelination. For reconstruction of entire lesion areas for the remyelination analysis, SEM was used. Images were acquired using the in-lens detector of a Merlin FEG SEM (Zeiss, Germany) operating at 2 KeV, attached to the ATLAS module (Zeiss, Germany). The entire lesion area was imaged as

multiple individual images, with overlap between adjacent ones. Alignment and merging of these images was performed using FIJI (ImageJ) and Photoshop CS5 or CS6 (Adobe). Brightness and contrast of EM images were adjusted for optimal detection of the structures.

## Antibodies

The following antibodies were used: Rabbit anti-cleaved Caspase 3 (IHC 1:500, Cat# 9661, Cell signaling Technology, Danvers, MA, USA; RRID:AB_2341188), mouse anti-CC1 (IHC 1:200 in the analysis of development, 1:300 in the analysis of remyelination, Cat# OP80, Merck Millipore, Billerica, MA, USA; RRID:AB_2057371), rabbit anti-FASN (IHC 1:200, Cat# Ab22759, Abcam, UK; RRID:AB_732316), chicken anti-GFP (IHC 1:1000, Cat# Ab13970, Abcam, UK; RRID:AB_300798), mouse anti-ki67 (IHC 1:200, Cat# m7249, clone MIB-5, Dako Agilent, Santa Clara, CA, USA; RRID:AB_2250503), rat anti-MBP (IHC 1:200 in the analysis of development, IHC 1:300 in the analysis of remyelination, Cat# MCA409S, Serotec/BioRad Laboratories, Hercules, CA, USA; RRID:AB_325004), goat anti-Olig2 (IHC 1:25, Cat# AF2418, R and D Systems, Minneapolis, MN, USA; RRID:AB_2157554; biotinylated), mouse anti-Olig2 (IHC 1:1000, clone 211F1.1, Cat# MABN50, Merck Millipore, Billerica, MA, USA; RRID:AB_10807410), rabbit anti-Olig2 (IHC 1:500 in the analysis of development, IHC 1:400 in the analysis of remyelination, Cat# AB9610, Merck Millipore, Billerica, MA, USA; RRID:AB_570666), rabbit anti-PDGFRα (IHC 1:500, Cat# 3174, Cell signaling Technology, Danvers, MA, USA; RRID:AB_2162345), donkey anti-chicken Alexa488 (IHC 1:1000, Cat# 703-545-155, Jackson ImmunoResearch, West Grove, PA, USA; RRID:AB_2340375), goat anti-mouse Alexa488 (IHC 1:300, Cat# A11029, Thermo Fisher Scientific, Waltham, MA, USA; RRID:AB_138404), donkey anti-mouse Alexa546 (IHC 1:1000, Cat# A10036, Thermo Fisher Scientific, Waltham, MA, USA; RRID:AB_2534012), goat anti-mouse Alexa546 (IHC 1:1000, Cat# A11030, Thermo Fisher Scientific, Waltham, MA, USA; RRID:AB_144695), goat anti-mouse Alexa647 (IHC 1:1000, Cat# A21237, Thermo Fisher Scientific, Waltham, MA, USA; RRID:AB_1500743), donkey anti-rabbit Alexa488 (IHC 1:1000, Cat# A21206, Thermo Fisher Scientific, Waltham, MA, USA; RRID:AB_2535792), donkey anti-rabbit Alexa647 (IHC 1:1000, Cat# A31573, Thermo Fisher Scientific, Waltham, MA, USA; RRID:AB_2536183), goat anti-rabbit Alexa546 (IHC 1:1000, Cat# A11035, Thermo Fisher Scientific, Waltham, MA, USA; RRID:AB_143051), goat anti-rabbit Alexa594 (IHC 1:300, Cat# A11012, Thermo Fisher Scientific, Waltham, MA, USA; RRID:AB_141359), goat anti-rabbit Alexa647 (IHC 1:1000, Cat# A21244, Thermo Fisher Scientific, Waltham, MA, USA; RRID:AB_141663), goat anti-rat Alexa647 (IHC 1:300, Cat# A21247, Thermo Fisher Scientific, Waltham, MA, USA; RRID:AB_141778), Streptavidin-Alexa647 (IHC 1:1000, Cat# S21374, Thermo Fisher Scientific, Waltham, MA, USA; RRID:AB_2336066).

## Immunostaining

Cell proliferation was analyzed using the Click-iT EdU Assay (Invitrogen), according to manufacturer's protocol. Mice received a single intraperitoneal injection of EdU (50 mg/Kg of weight) and the tissue of interest was harvested 2 hr after the injection.

Mice were anaesthetized by terminal intraperitoneal injection of pentobarbital, 10% in saline solution (0.9% NaCl) (Eskonarkon, Streuli Pharma, Switzerland). Mice were intracardially perfused with 1x PBS pH 7.4, followed by 4% paraformaldehyde, 5% sucrose in 1x PBS. Dissected tissues were postfixed in 4% paraformaldehyde, 5% sucrose in 1x PBS overnight at 4°C and cryoprotected in 30% sucrose in PBS for 24 hr at 4°C. Tissues embedded in O.C.T. Tissue Tek (Sakura, The Netherlands) were cut on a cryostat in 10 μm sections with distance between serial sections of 100 μm, and transferred to SuperFrost Plus (Thermo Fisher Scientific, Waltham, MA, USA) coated slides. Slides were stored at −80°C until further use.

For immunostaining, slides were defrosted at room temperature (RT) for at least 30 min and rinsed 3x with 1% Triton X-100 in 1x PBS. Blocking and permeabilization was done with 1% Triton X-100, 10% goat or donkey serum, in 1x PBS for 30 min. Sections were incubated with primary antibodies, diluted in blocking solution, overnight at 4°C. The following day, sections were washed 3 × 10 min with 0.1% Triton X-100 in 1x PBS and incubated with secondary antibodies, diluted in blocking solution, for 2 hr at RT. Sections were washed twice with 0.1% Triton X-100 in 1x PBS, and incubated with 4′,6′-diamidino-2-phenylindole (DAPI) for 10 min. After a last 10 min wash with 1x PBS, sections were coverslipped with ImmuMount (Thermo Fisher Scientific, Waltham, MA, USA). For CC1, cleaved caspase 3, and Ki67 stainings, an antigen retrieval step was added to the above

protocol. Briefly, after thawing slides were incubated in a pre-heated antigen retrieval solution in a water bath for 20 min (for analysis of development: Sodium citrate buffer (10 mM Tri-sodium citrate dehydrate, 0.05% Tween-20, pH 6.0) incubated at 95℃, for analysis of remyelination Tris-EDTA buffer (10 mM Trizma Base (Sigma-aldrich, St Louis, MI, USA), 1 mM EDTA, 0.05% Tween-20, pH 9.0) incubated at 86℃). For stainings with mouse anti-CC1 antibody performed for the analysis of remyelination, sections were next incubated with mouse blocking reagent (Cat# MKB-2213, Vector Laboratories, Burlingame, CA, USA) for 1 hr at RT followed by primary antibody incubation and further processed according to manufacturer's recommendations in 1% Triton X-100 in 1x PBS to prevent cross-reaction of anti-mouse secondary antibodies with endogenous antibodies.

For stainings with biotinylated anti-Olig2 antibody, sections were treated with a Streptavidin/Biotin blocking Kit (Cat# SP-2002, Vector Laboratories, Burlingame, CA, USA) according to the manufacturer's protocol.

## Lipid analysis

For lipidomic analysis, myelin from spinal cords dissected from P60 control and mutant mice was isolated by sucrose density gradient centrifugation, as described previously (*Larocca and Norton, 2007*). Briefly, spinal cords were transferred in suspension buffer (0.3 M sucrose containing as adjuvants 20 mM Tris-HCl pH 7.4, 1 mM EDTA, 1 mM DTT, 100 µM phenylmethylsulfonyl fluoride (PMSF), 10 µg/ml leupeptin, and 10 µg/ml antipain) and tissue disrupted in a glass and stainless steel homogenizer. The homogenate was layered onto a 0.83 M sucrose solution with the same adjuvants, and centrifuged at 75000 g for 30 min at RT. The obtained white band of crude myelin membranes was collected and subjected to three cycles of osmotic shock in 20 mM Tris-HCl pH 7.4 with adjuvants, 1x at 75000 g and 2x at 12000 g at 4℃. Myelin pellets were resuspended in suspension buffer and subjected to a repetition of density centrifugation and osmotic shock. Finally, obtained myelin pellets were resuspended in 0.83 M sucrose solution with adjuvants. This was overlaid with suspension buffer and centrifuged at 75000 g for 30 min at 4℃. The myelin fraction was subjected to a last repetition of osmotic shock and the pellet resuspended in Tris-HCl buffer and immediately frozen in liquid nitrogen. Purified myelin was stored at −80℃ until further processing.

Lipid standards were obtained from Avanti Polar Lipids (Alabaster, AL, USA). Analysis of lipid species was performed according to published methods (*Fauland et al., 2011*; *Triebl et al., 2017*). Samples were extracted with methyl tert-butyl ether (MTBE) (*Matyash et al., 2008*). Lipid extracts were evaporated and resuspended in 1 ml chloroform/methanol (1/1; v/v). Each lipid extract was then split for analysis of total fatty acids (350 µl), positive ESI LC-MS/MS (18 µl) and negative ESI LC-MS/MS (18 µl). Lipid extracts for LC-MS/MS analysis were evaporated, spiked with a mix of quantitative LIPID MAPS internal standards and 2 µl of spiked samples were injected onto a Thermo 1.9 µm Hypersil GOLD C8, 100 × 1 mm HPLC column mounted in an Dionex Ultimate 3000 UHPLC instrument (Thermo Fisher Scientific). Data acquisition was performed by Orbitrap Velos Pro (Thermo Fisher Scientific) full scans at a resolution of 100 k and <5 ppm mass accuracy with external calibration. Nominal mass resolution product ion spectra were acquired in preview mode at the LTQ. A chromatography with Electron Impact Mass Spectrometry (GC-EI/MS) of Total fatty acids (free + esterified) were determined by gas chromatography - electron impact mass spectrometry (GC-EI/MS). Lipid extracts were dried and suspended in 1 ml methanolic NaOH. After 10 min incubation at 80℃, samples were cooled for 5 min on ice. Then, 1 ml $BF_3$ was added and incubated for 10 min at 80℃. Fatty acid methyl esters were extracted with 1 ml saturated NaCl and 2 ml hexane. The hexane phase was dried and methyl esters dissolved in 1.5 ml hexane. A Trace-DSQ GC-MS (Thermo Fisher Scientific) equipped with a 30 m column (model TR-FAME, Thermo Fisher Scientific) was used for analysis. Helium was used as carrier gas at a flow of 1.3 ml/min, in split mode, at 250℃ injector temperature. Initial oven temperature of 150℃ was held for 0.5 min and then temperature was increased to 180℃ at a rate of 10 ℃/min. This was followed by a further increase to 190℃ at a rate of 0.5 ℃/min and then increased to 250℃ at a rate of 40 ℃/min and kept for 3 min. The mass spectrometer was run in EI mode and fatty acids were detected in full scan of m/z 80–400. Source temperature was set to 250℃ and the transfer line temperature to 200℃. Internal standards used in LC-MS were (product number, lipid shorthand nomenclature, amount/sample): LM-1100, PE 12:0/13:0, 160 pmol; LM-1102, PE 17:0/20:4, 160 pmol; LM-1103, PE 21:0/22:6, 160 pmol; LM-1104, PE 17:0/14:1, 160 pmol; LM-1302, PS 17:0/20:4, 240 pmol; LM-1300, PS 12:0/13:0, 240 pmol; LM-1304, PS 17:0/14:1, 240 pmol; LM-1000, PC 12:0/13:0, 200 pmol; LM-1002, PC 17:0/20:4, 200 pmol; LM-

1003, PC 21:0/22:6, 200 pmol; LM-1004, PC 17:0/14:1, 200 pmol; LM-1601, LPC 17:1, 80 pmol; LM-6002, sphingolipid mix, 120 pmol; LM-1500, PI 12:0/13:0, 320 pmol; LM-1502, PI 17:0/20:4, 320 pmol; LM-1504, PI 17:0/14:1, 320 pmol.

LC = liquid chromatography, ESI = electrospray ionization, MS = mass spectrometry, FT-ICR = Fourier transform ion cyclotron resonance, GC = gas chromatography, EI = electron impact, DSQ = dual stage quadrupole, m/z = mass/ charge number, LM-n = Lipid Maps-ID number, PE = phosphatidylethanolamine, PS = phosphatidylserine, PC = phosphatidylcholine, PI = phosphatidylinositol.

Data were processed by Lipid Data Analyzer software as previously described (*Hartler et al., 2017*), relying on exact mass, retention time and MS/MS spectra. Annotation of lipid species followed the shorthand nomenclature of the International Lipid Classification and Nomenclature Committee (*Liebisch et al., 2013*). During lipid extraction procedure, the fraction containing proteins was lyophilized. Proteins were dissolved in lysis buffer (1% Rapigest (Waters/Millipore, Billerica, MA, USA), Tris pH 7.4, 0.1 mM EDTA) and quantified on a Nanodrop Lite (Nanodrop Technologies, Thermo Fisher Scientific) spectrophotometer. Lipid amounts were normalized to the sample total protein content.

## Sequencing

Total RNA was isolated with a Qiagen MiniKit according to manufacturer's protocols, quantity and quality were determined with Nanodrop Lite (NanoDrop Technologies, Thermo Fisher Scientific), Qubit 2.0 Fluorometer (Life Technologies, Carlsbad, CA, USA) and Bioanalyzer 2100 (Agilent, Santa Clara, CA). The TruSeq Stranded mRNA Sample Prep Kit (Illumina, San Diego, CA, USA) was used in the succeeding steps. Briefly, poly-A containing RNA molecules were purified from the total RNA samples (100 ng) using oligo-dT attached magnetic beads. Isolated RNA was reverse-transcribed into double-stranded cDNA, with actinomycin added during first-strand synthesis. The cDNA samples were fragmented, end repaired and polyadenylated before ligation of TruSeq adapters. Fragments containing TruSeq adapters on both ends were selectively enriched with PCR. The quality and quantity of the enriched libraries were validated using Qubit 2.0 Fluorometer and Bioanalyzer 2100. The product is a smear with an average fragment size of approximately 360 bp. The libraries were normalized to 10 nM in Tris-Cl 10 mM, pH 8.5 with 0.1% Tween-20. Sequencing was performed on an Illumina HiSeq 4000 in single end 125 bp mode. The raw reads were first cleaned by removing adapter sequences, trimming low quality ends, and filtering reads with low quality (phred quality <20). Read-alignment was done with STAR (v2.5.1b). Ensembl genome build GRCm38 was used as reference and gene annotations downloaded on 2015-06-25 from Ensembl. Applied STAR alignment options were: OutFilterType BySJout, outFilter MatchNmin30, outFilter Mismatch Nmax10, outFilter MismatchNoverLmax0.05, alignSJDB overhangMin1, aligns overhangMin8, alignIntron Max1000000, alignMatesGap Max1000000, outFilterMultimapN max50. Quantification of gene level expression was carried out using RSEM (v1.2.22) (*Li and Dewey, 2011*). Differential expression was computed using the generalized linear model implemented in the Bioconductor package EdgeR (*Robinson et al., 2010*). To correct for multiple testing, the Benjamini-Hochberg algorithm was applied and adjusted p-values computed. Only significantly differentially expressed transcripts (FDR ≤ 0.05, no fold change threshold) were loaded onto Metacore (vs6.33) for enrichment analysis of GeneOntology processes and Metacore pathways.

## qRT-PCR

Total RNA was isolated with a Qiagen MiniKit according to the manufacturer's protocols; quantity and quality were determined with Nanodrop Lite (NanoDrop Technologies). RNA samples were reverse-transcribed to cDNA with oligo(dT) random primers in the presence of RNaseOUT (Thermo Fisher Scientific, Waltham, MA, USA) with the Maxima RT kit (Thermo Fisher Scientific, Waltham, MA, USA). Targeted sequences were amplified with exon/exon boundary-spanning probes and detected by measurement of SYBRgreen on a LightCycler 480 (Roche, Switzerland). Cp values were determined with the LightCycler 480 software (Roche, Switzerland). Primers used were: SOX10 Forward 5'-CCGACCAGTACCCTCACCT-3', Reverse 5'-TCAATGAAGGGGCGCTTGT-3', MYRF Forward 5'-ATGGAGGTGGTGGACGAGAC-3', Reverse 5'-GGCGTCCTCTTTGCCAATGT-3', MBP Forward 5'-

ACACGAGAACTACCCATTATGGC-3', Reverse 5'-CCAGCTAAATCTGCTGAGGGGA-3', Actin Forward 5'-GTCCACACCCGCCACC-3', Reverse 5'-GGCCTCGTCACCCACATAG-3'.

## Statistical analysis

All experiments were quantified blindly to the genotype and treatment group. Statistics were analyzed using GraphPad Prism vs6.01. Data were assumed to be normally distributed, but not formally tested. Variance was assumed to be equal between groups of data. No statistical methods were used to predetermine sample size, but our sample sizes are similar than those generally employed in the field. Statistical significance was determined using an unpaired two-tailed two sample Student's t-test for two group comparisons. Multiple groups analysis was performed with analysis of variance (ANOVA) and post-hoc test as detailed in text and figures. Data show mean ±SEM. Significance was set at $p < 0.05$ *, $p < 0.01$ **, $p < 0.001$ ***.

## Data availability

RNA-sequencing data are available on NCBI's GEO database, accession number GSE112725.

## Acknowledgements

We thank all members of the Suter lab for data discussion, the Functional Genomic Center of Zürich and the ScopeM imaging facility of ETH Zürich for excellent technical support. We also thank Drs. William Richardson and David Rowitch for transgenic mice and advice. This work was supported by the Swiss National Science Foundation (SNF) to US. LM was supported by a SNF Marie-Heim-Voegtlin fellowship (PMPDP3 139610).

## Additional information

### Funding

| Funder | Grant reference number | Author |
|---|---|---|
| Schweizerischer Nationalfonds zur Förderung der Wissenschaftlichen Forschung | | Ueli Suter |
| Schweizerischer Nationalfonds zur Förderung der Wissenschaftlichen Forschung | PMPDP3 139610 | Laura Montani |

The funders had no role in study design, data collection and interpretation, or the decision to submit the work for publication.

### Author contributions

Penelope Dimas, Conceptualization, Data curation, Formal analysis, Investigation, Visualization, Methodology, Writing—original draft, Writing—review and editing; Laura Montani, Conceptualization, Data curation, Formal analysis, Supervision, Funding acquisition, Investigation, Visualization, Methodology, Writing—original draft, Writing—review and editing; Jorge A Pereira, Conceptualization, Data curation, Formal analysis, Validation, Investigation, Visualization, Methodology, Writing—review and editing; Daniel Moreno, Data curation, Formal analysis, Investigation, Methodology, Writing—review and editing; Martin Trötzmüller, Conceptualization, Data curation, Formal analysis, Investigation, Methodology, Writing—review and editing; Joanne Gerber, Data curation, Formal analysis, Investigation, Visualization, Methodology, Writing—review and editing; Clay F Semenkovich, Resources, Methodology, Writing—review and editing; Harald C Köfeler, Conceptualization, Resources, Data curation, Formal analysis, Investigation, Methodology, Writing—review and editing; Ueli Suter, Conceptualization, Resources, Data curation, Formal analysis, Supervision, Funding acquisition, Project administration, Writing—review and editing

Author ORCIDs
Laura Montani (iD) https://orcid.org/0000-0002-8243-6734
Jorge A Pereira (iD) http://orcid.org/0000-0002-0159-4133
Ueli Suter (iD) https://orcid.org/0000-0002-9211-5184

## Ethics

Animal experimentation: All animal experiments were performed with the approval and in strict accordance with the guidelines of the Zurich Cantonal Veterinary Office (Permit Numbers ZH161/2014, ZH090/2017, ZH03/2012, ZH264/2014, ZH207/2017).

## Decision letter and Author response

Decision letter https://doi.org/10.7554/eLife.44702.029
Author response https://doi.org/10.7554/eLife.44702.030

# Additional files

## Supplementary files

• Transparent reporting form
DOI: https://doi.org/10.7554/eLife.44702.025

## Data availability

Sequencing data have been deposited in GEO under accession code GSE112725. All data generated or analysed during this study are included in the manuscript and supporting files. Source data files have been provided for Figure 5.

The following dataset was generated:

| Author(s) | Year | Dataset title | Dataset URL | Database and Identifier |
|---|---|---|---|---|
| Montani L | 2018 | Role of fatty acid synthase in oligodendrocyte myelination | https://www.ncbi.nlm.nih.gov/geo/query/acc.cgi?acc=GSE112725 | NCBI Gene Expression Omnibus, GSE112725 |

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
