## [Decision Letter]

Thank you for submitting your article "CNS myelination and remyelination depend on fatty acid synthesis by oligodendrocytes" for consideration by *eLife*. Your article has been reviewed by Marianne Bronner as the Senior Editor, a Reviewing Editor, and two reviewers. The following individuals involved in review of your submission have agreed to reveal their identity: Teresa Wood (Reviewer #2).

The reviewers have discussed the reviews with one another and the Reviewing Editor has drafted this decision to help you prepare a revised submission.

Summary:

This study explored the role of fatty acid synthesis in the development of oligodendrocytes and myelin during early postnatal life in mice, and remyelination in the adult CNS. The authors conditionally deleted fatty acid synthase (FASN) from oligodendrocyte progenitors (OPCs) in the developing CNS and found that while this did not impair the proliferation and differentiation of OPCs, it slowed the process of myelination, altered the lipid composition of myelin and disrupted myelin integrity. This delay in myelination was partially reversed by feeding FASN cKO mice a high fat diet. Deletion of FASN from OPCs in the adult CNS similarly impaired the regeneration of myelin sheaths. Together, these findings indicate that de novo fatty acid synthesis is critical for both developmental myelination and remyelination.

Essential revisions:

1) The authors should quantify the percentage recombination and knockout of FASN in both their developmental model and tamoxifen inducible model. They have the tools to do so and may even have the data to hand in tissue collected. This is particularly important in addressing the apparent diversity in phenotype between optic nerve and spinal cord, and in the difference in phenotypic severity of developmental myelination and remyelination in the spinal cord.

Could such differences be explained simply by recombination efficiency?

If recombination efficiency is similar in spinal cord and optic nerve, the authors should still discuss whether the difference they see there in developmental myelination really reflects intrinsic differences in oligodendrocytes, as suggested, or possibly local differences in astrocyte coupling etc.

2) Figure 3: EM analysis shows an increase in unmyelinated axons in mutant mice. The authors conclude that FASN is critical for timely onset of myelination. However, the decrease in myelinated axons may not be due to timing of onset, especially since recovery does not occur at P180. Since oligodendrocytes wrap multiple axons per cell, the phenotype could also be attributed to less myelin being produced by each mutant oligodendrocyte and/or fewer myelin sheaths. Another possibility is that the aberrant myelin observed in the mutants reduces their ability to wrap the same number of axon segments. Wording should be altered to reflect other possible conclusions from the EM data.

3) Figure 3—figure supplement 1: The authors address possible differential dependence on FASN by studying myelin in the spinal cord and optic nerve and conclude that FASN is critical in diverse OL populations. Do the authors have any information from their existing samples on the corpus callosum to strengthen this conclusion?

4) Figure 8: It is unclear whether the decrease in CC1+ cells at 14 and 21 dpl is due to a deficit in the OPC population, a differentiation deficit, or cell death. TUNEL staining may capture more dying cells than cC3 staining and should be paired with a marker for OPCs and a marker for mature OLs to determine which population is affected. Some information on this could be accomplished on existing tissues. The authors should try to increase their "n" for the experiment assessing the contribution of cell death to the phenotypes observed in oligodendrocyte number in mutants during remyelination. It is great that the authors show actual data points, but this reveals huge diversity in Figure 8—figure supplement 3D and E, and I strongly suspect that this experiment would not have had sufficient power to detect a statistically significant increase in cell death, which is indeed suggested by the large fold increase. I appreciate that achieving high statistical power in mammalian in vivo experiments is challenging, but I think this issue needs to be acknowledged. At the very least the authors should discuss this point, lest they run the risk of conveying a potentially misleading false negative.

5) Figure 8: The observed reduction in recombined OPCs in FASN mutants during remyelination (subsection “Increased dietary intake of lipids can partially rescue lack of fatty acid synthesis by oligodendrocytes”) needs to be further discussed. It is interesting that developmental OPCs are unaffected, but adult OPCs show a deficit during remyelination.

6) Is it possible for the authors to try a high fat diet supplementation following remyelination? This could help assess whether lipid synthesis is a bottleneck to remyelination. Apologies if this has been tested in the past by others, but, if not, this could be a timely opportunity to do so. The authors discuss their surprise in observing a strong phenotype observed for remyelination in the FASN inducible mutants. However, it is possible that the dietary support, or shuttling through astrocytes is actually compromised by the environment of a demyelinating lesion, and it would be interesting to see if high fat diet could help circumvent any such disruption.

---

## [Author Response]

Essential revisions:1) The authors should quantify the percentage recombination and knockout of FASN in both their developmental model and tamoxifen inducible model. They have the tools to do so and may even have the data to hand in tissue collected. This is particularly important in addressing the apparent diversity in phenotype between optic nerve and spinal cord, and in the difference in phenotypic severity of developmental myelination and remyelination in the spinal cord.Could such differences be explained simply by recombination efficiency?If recombination efficiency is similar in spinal cord and optic nerve, the authors should still discuss whether the difference they see there in developmental myelination really reflects intrinsic differences in oligodendrocytes, as suggested, or possibly local differences in astrocyte coupling etc.

We have now quantified the recombination percentages using a Cre-dependent reporter allele (since FASN protein was barely detectable in oligodendrocyte progenitor cells). The corresponding data have been added to the revised manuscript (Figure 1—figure supplement 1, Figure 3—figure supplement 1, and Figure 7—figure supplement 1). Accordingly, we have also adjusted the text. However, we have refrained from making potentially over-interpreting statements based on our findings. This choice relates to the highly relevant points raised by the reviewers about potential oligodendrocyte-intrinsic versus local tissue differences in the various regions. Indeed, our evaluation of various myelinated regions was intended to examine whether FASN expression is in principle required for anatomically differently located oligodendrocytes. Thus, the main conclusion of our analysis is that accurate developmental myelination in the spinal cord, the optic nerve, and in the corpus callosum (see answer to point 3 below) depends on FASN expressed by oligodendrocytes. To which extent our findings in the different regions might relate to different environments and physiological contexts, or might be due to potential intrinsic differences between oligodendrocyte lineage cells, remains to be investigated. We have altered the wording in the revised manuscript to reflect these points more clearly. We thank the reviewers for indicating this potential misunderstanding out to us.

2) Figure 3: EM analysis shows an increase in unmyelinated axons in mutant mice. The authors conclude that FASN is critical for timely onset of myelination. However, the decrease in myelinated axons may not be due to timing of onset, especially since recovery does not occur at P180. Since oligodendrocytes wrap multiple axons per cell, the phenotype could also be attributed to less myelin being produced by each mutant oligodendrocyte and/or fewer myelin sheaths. Another possibility is that the aberrant myelin observed in the mutants reduces their ability to wrap the same number of axon segments. Wording should be altered to reflect other possible conclusions from the EM data.

We agree that our wording in this context was not optimal. Thus, we have ameliorated the text to provide a more precise description of the various possibilities, following the valuable suggestions of the reviewers. We thank the reviewers for this input.

3) Figure 3—figure supplement 1: The authors address possible differential dependence on FASN by studying myelin in the spinal cord and optic nerve and conclude that FASN is critical in diverse OL populations. Do the authors have any information from their existing samples on the corpus callosum to strengthen this conclusion?

We have now analyzed the corpus callosum. The corresponding results, revealing that correct myelination in this region is also dependent on FASN expression by oligodendrocytes, have been added to the manuscript (Figure 3—figure supplement 3, and text).

4) Figure 8: It is unclear whether the decrease in CC1+ cells at 14 and 21 dpl is due to a deficit in the OPC population, a differentiation deficit, or cell death. TUNEL staining may capture more dying cells than cC3 staining and should be paired with a marker for OPCs and a marker for mature OLs to determine which population is affected. Some information on this could be accomplished on existing tissues. The authors should try to increase their "n" for the experiment assessing the contribution of cell death to the phenotypes observed in oligodendrocyte number in mutants during remyelination. It is great that the authors show actual data points, but this reveals huge diversity in Figure 8—figure supplement 3D and E, and I strongly suspect that this experiment would not have had sufficient power to detect a statistically significant increase in cell death, which is indeed suggested by the large fold increase. I appreciate that achieving high statistical power in mammalian in vivo experiments is challenging, but I think this issue needs to be acknowledged. At the very least the authors should discuss this point, lest they run the risk of conveying a potentially misleading false negative.

We have dedicated substantial efforts to characterize cell death in the oligodendrocyte lineage. Beyond the data included in the manuscript, which consists of the percentage of CC1+ YFP+ cC3+ oligodendrocytes at 11 dpl, we have also assessed:

a) The population of CC1- YFP+ cC3+ not-differentiated oligodendrocytes and found only one positive cell in one of the animals analyzed (quantified on the same images which generated the data included in the manuscript, at 11 dpl).

b) Additional time points (10 dpI, 12 dpl; analysis of sections covering the entire lesions in 3 control and 3 i-cMU mice) revealed no YFP+ Cc3+ cells.

We also tested three different TUNEL protocols and assay kits from two independent manufacturer's, but unfortunately, none of these methods enabled, in our hands, robust co-staining of TUNEL with antibodies recognizing either YFP or CC1. As the careful comparative analysis of cell death in controls versus mutants mandates the colocalization of apoptosis markers with oligodendrocyte lineage cells and a recombination marker, we are unable to draw firm conclusions from these experiments.

Despite our best efforts, and also due to technical limitations of the experiments and available tools, we consider it unlikely that extending the analysis to increase “n” would provide a satisfying answer in this context. However, as suggested by the reviewer, we acknowledge now this limitation of our analysis explicitly in the revised manuscript.

5) Figure 8: The observed reduction in recombined OPCs in FASN mutants during remyelination (subsection “Increased dietary intake of lipids can partially rescue lack of fatty acid synthesis by oligodendrocytes”) needs to be further discussed. It is interesting that developmental OPCs are unaffected, but adult OPCs show a deficit during remyelination.

We discuss this issue now in the revised manuscript in condensed form. One interpretation of our findings is that the observed decrease in CC1+ recombined OLs in mutant animals during remyelination promotes a mild reaction from the aOPCs, potentially with the goal of some compensation. If the recombined cells cannot be sustained and die while differentiating from aOPCs towards myelinating oligodendrocytes, this may cause a slight reduction of the remaining numbers of recombined aOPCs. Alternatively, aOPCs may have intrinsic differences in their dependency on FASN compared to developmental OPCs with regard to survival. We consider this less likely given the low expression of FASN by developmental OPCs and aOPCs.

6) Is it possible for the authors to try a high fat diet supplementation following remyelination? This could help assess whether lipid synthesis is a bottleneck to remyelination. Apologies if this has been tested in the past by others, but, if not, this could be a timely opportunity to do so. The authors discuss their surprise in observing a strong phenotype observed for remyelination in the FASN inducible mutants. However, it is possible that the dietary support, or shuttling through astrocytes is actually compromised by the environment of a demyelinating lesion, and it would be interesting to see if high fat diet could help circumvent any such disruption.

We agree with the reviewers that such studies are highly interesting, including investigations of the conceptual issue of lipid flux from the diet to oligodendrocytes in the context of remyelinating lesions, taking into account shuttling through astrocytes. However, such a study is quite challenging to perform accurately. Mainly due to the complexity of the lesion setting (including potential damage to the blood-CNS barrier), the accurate selection of the experimental conditions and the interpretation of the study will be demanding, not the least since high fat diet supplementation is likely to influence inflammation and potentially other variables. Thus, we feel that such an analysis requires a rather extensive dedicated research project on its own to yield comprehensive and robust results.